# nNOS-expressing neurons in the vmPFC transform pPVT-derived chronic pain signals into anxiety behaviors

Hai-Ying Liang[1,6,7], Zhi-Jin Chen[1,7], Hui Xiao[1], Yu-Hui Lin [1], Ying-Yi Hu[1], Lei Chang[1], Hai-Yin Wu[1,2,3], Peng Wang[4], Wei Lu [5], Dong-Ya Zhu[1,2,3] & Chun-Xia Luo [1,2,3 ✉]

Anxiety is common in patients suffering from chronic pain. Here, we report anxiety-like behaviors in mouse models of chronic pain and reveal that nNOS-expressing neurons in ventromedial prefrontal cortex (vmPFC) are essential for pain-induced anxiety but not algesia, using optogenetic and chemogenetic strategies. Additionally, we determined that excitatory projections from the posterior subregion of paraventricular thalamic nucleus (pPVT) provide a neuronal input that drives the activation of vmPFC nNOS-expressing neurons in our chronic pain models. Our results suggest that the pain signal becomes an anxiety signal after activation of vmPFC nNOS-expressing neurons, which causes subsequent release of nitric oxide (NO). Finally, we show that the downstream molecular mechanisms of NO likely involve enhanced glutamate transmission in vmPFC CaMKIIα-expressing neurons through S-nitrosylation-induced AMPAR trafficking. Overall, our data suggest that pPVT excitatory neurons drive chronic pain-induced anxiety through activation of vmPFC nNOS-expressing neurons, resulting in NO-mediated AMPAR trafficking in vmPFC pyramidal neurons.

[1] Department of Pharmacology, School of Pharmacy, Nanjing Medical University, Nanjing 211166, China. [2] Collaborative Innovation Center for Cardiovascular Disease Translational Medicine, Nanjing Medical University, Nanjing 211166, China. [3] Guangdong-Hong Kong-Macao Greater Bay Area Center for Brain Science and Brain-Inspired Intelligence, Guangzhou 510515, China. [4] State Key Laboratory of Translational Medicine and Innovative Drug Development, Jiangsu Simcere Pharmaceutical Co. Ltd., Nanjing 210042, China. [5] State Key Laboratory of Bioelectronics, The MOE Key Laboratory of Developmental Genes and Human Disease, Institute of Life Sciences, Southeast University, Nanjing 210096, China. [6]Present address: The First Affiliated Hospital of Fujian Medical University, Longyan 364000, China. [7]These authors contributed equally: Hai-Ying Liang and Zhi-Jin Chen. ✉email: chunxialuo@njmu.edu.cn

Epidemiological studies have demonstrated that psychiatric disorders, such as anxiety, are common in patients suffering from chronic pain[1–3]. Chronic pain and anxiety not only are common comorbidities but also contribute to each other's development[4–6]. Despite these consistent epidemiological data, previous basic studies on pain and anxiety were usually performed independently[7]. For pain, researchers have mainly focused on the discovery of peripheral nociceptors and the gated control of pain transmission along the spinal cord[7,8]. In contrast, anxiety has been previously investigated primarily from the aspect of psychology[9,10]. The specific neural circuits and molecular signals implicated in chronic pain-induced anxiety are still not fully understood, impeding appropriate therapeutic strategies from being adopted.

Many brain areas are reported to be involved in the regulation of anxiety, and some of them are also associated with pain, such as the anterior cingulate cortex (ACC), ventromedial prefrontal cortex (vmPFC), insular cortex, and amygdala[7,10,11]. The common brain regions provide the structure base for chronic pain to induce anxiety. Recently, studies on excitatory transmission at the synaptic level in the ACC have revealed that two forms of long-term potentiation (LTP), the kainite receptor-dependent pre-LTP and the N-methyl-D-aspartic acid receptor (NMDAR)-dependent post-LTP, contribute to interactions between chronic pain and anxiety[12,13]. In addition to the ACC, another important area for the interaction between chronic pain and anxiety is the vmPFC[14], which is considered by some researchers to comprise the prelimbic cortex (PL) and infralimbic cortex (IL)[15–17]. The role of the vmPFC in anxiety regulation has been widely investigated. Though functional imaging studies indicate that the mPFC is hypoactive in anxiety patients[18], subjects with the highest levels of anxiety tended to exhibit the smallest decreases in vmPFC activity[19]. Findings from classic pharmacological studies indicate that stimulation of NMDARs in PL[20] or α-amino-3-hydroxy-5-methyl-4-isoxazolepropionic acid receptors (AMPARs) in IL[21] induces anxiety-like behaviors, whereas the activation of gamma-amino butyric acid (GABA)$_A$ receptors in the vmPFC produces an anxiolytic-like response[22]. Moreover, the vmPFC is involved in pain perception and integration, as rats with chronic pain show morphological and functional reorganization of the vmPFC[23] and increased glutamate release in the vmPFC[24], and patients with persistent chronic pain have greater mPFC activity[25]. Therefore, we hypothesize that the vmPFC is pivotal in chronic pain-induced anxiety, in addition to the ACC. However, few studies have focused on this brain region in investigations into chronic pain-induced anxiety[26], so further research is needed.

In our previous studies, we demonstrated that neuronal nitric oxide synthase (nNOS) and its product nitric oxide (NO) are critical for mediating anxiety-related behaviors in mice. Inhibition or knockout of hippocampal nNOS produces anxiolytic effects[27], and coupling of nNOS with its partner protein CAPON (carboxy-terminal PDZ ligand of nNOS) is a potential strategy for developing new anxiolytics[28]. There are many nNOS-expressing neurons scattered throughout the neocortex[29], including the vmPFC. Increased nNOS enzyme activity and NO production in the vmPFC are associated with the long-lasting anxiogenic-like effect induced by predator exposure[30] and acute restraint stress[31]. Moreover, NO is an important neurotransmitter involved in the nociceptive process, and it contributes to the development of central sensitization; further, nNOS inhibition can considerably reduce both inflammatory and neuropathic pain[32,33]. Therefore, we hypothesize that nNOS-expressing neurons and the nNOS–NO signaling pathway in the vmPFC are involved in chronic pain-induced anxiety.

In this study, we selectively manipulate the activity of nNOS-expressing neurons in the vmPFC to investigate the role of the vmPFC in chronic pain-induced anxiety and the contribution of the nNOS–NO signaling pathway after nNOS-expressing neuron stimulation. Then, we determine the excitatory projections from the posterior subregion of paraventricular thalamic nucleus (pPVT) to the vmPFC as the neural circuit controlling nNOS-expressing neurons of the vmPFC during chronic pain. Our findings indicate that the pain signal is transformed into an anxiety signal after nNOS-expressing neurons in the vmPFC were activated. Additionally, we explore the downstream molecular mechanisms following NO production and propose a local activity regulation in the vmPFC mediating chronic pain-induced anxiety, which is associated with enhanced AMPAR trafficking and function through S-nitrosylation.

## Results

**The role of vmPFC nNOS-expressing neurons.** Chronic inflammatory pain was induced in mice by treatment with complete Freund's adjuvant (CFA)[13,26], and von Frey monofilaments were used to assess mechanical hyperalgesia[13,34]. Though the mice exhibited a 50% decrease in paw withdrawal threshold just 1 h after CFA injection (Supplementary Fig. 1a), they did not show anxiety-like behaviors (Supplementary Fig. 1b, c). Three days after CFA injection, mechanical hyperalgesia (Fig. 1a) was observed, and it was accompanied by decreased distance in the center in the open field (OF) test (Fig. 1b) and reduced time in the open arms in the elevated plus maze (EPM) test (Fig. 1c), indicating that anxiety-like behaviors were induced by chronic pain. CFA-injected mice exhibited locomotor activities similar to those of normal saline (NS)-injected mice (Fig. 1b, c). Moreover, time course experiments showed that mechanical hyperalgesia and anxiety-like behaviors were persistent until 20 days after CFA injection (Supplementary Fig. 1d–i). At day 28, the anxiety-like behaviors disappeared along with chronic pain (Supplementary Fig. 1j–l).

As shown in Fig. 1d–f, compared with NS-injected mice, CFA-injected mice had more nNOS-expressing neurons co-labeled with c-Fos (a widely used indicator of cell activation) as well as total c-Fos$^+$ cells in the vmPFC at 4 h after injection. nNOS-expressing neurons are classically divided into two types based on nNOS immunofluorescence: type I (strongly labeled) and type II (weakly labeled). CFA-induced neuron activation occurred in both types of nNOS-expressing neurons (Supplementary Fig. 2). To determine whether vmPFC nNOS-expressing neurons were indeed involved in the control of anxiety-like behaviors induced by chronic pain, we specifically manipulated the activity of this neuron population through chemogenetics. First, we expressed hM3Dq, a clozapine N-oxide (CNO)-based excitatory designer receptor exclusively activated by designer drugs (DREADD), using an adeno-associated virus (AAV-hSyn-DIO-hM3Dq-eGFP) in the vmPFC of nNOS-Cre mice (Fig. 1g). Three weeks later, eGFP fluorescence verified the accurate expression of hM3Dq (Fig. 1h). hM3Dq-eGFP was expressed in both type I (marked by arrows, approximately 10%) and type II (marked by arrowheads, approximately 90%) nNOS-expressing neurons (Fig. 1i). Cre-dependent expression in type II nNOS neurons was also confirmed using AAV-hSyn-DIO-mCherry-infected nNOS-Cre mice (Supplementary Fig. 3a, b). Moreover, fluorescence in situ hybridization (FISH) also indicated that Cre-dependent mRNA expression was specific to nNOS neurons (Supplementary Fig. 3c, d). Increased action potential (AP) firing of hM3Dq-eGFP-expressing neurons after CNO application to brain slices indicated the efficient stimulation of these neurons (Fig. 1j). Behavioral tests performed 30 min after CNO or NS microinjection showed that chemogenetic activation of nNOS-expressing neurons in the vmPFC caused obvious anxiety-like behaviors; this was revealed by both

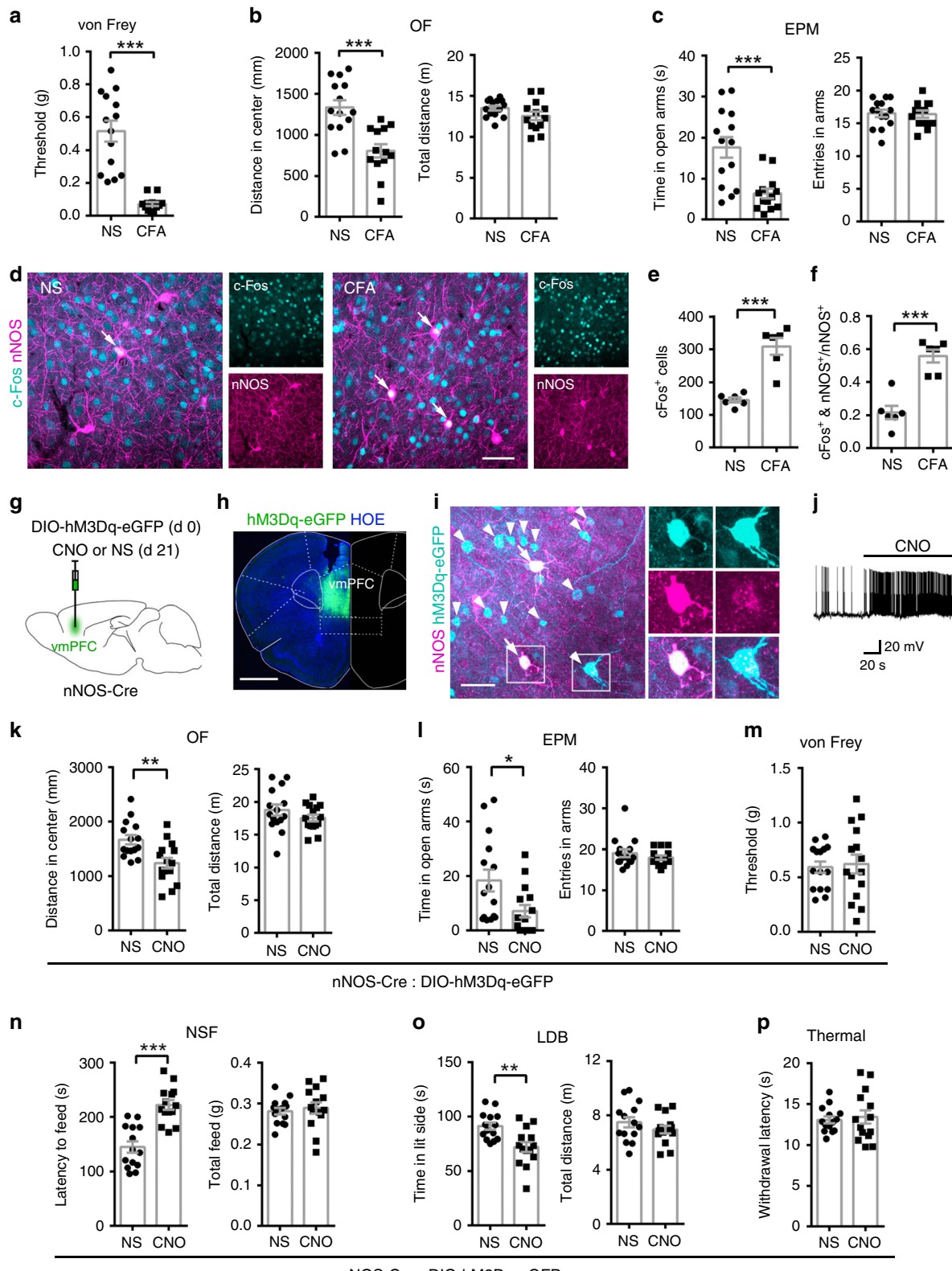

the OF test (Fig. 1k) and EPM test (Fig. 1l), but there was no influence on locomotor activities (Fig. 1k, l). We also confirmed the above behavioral effects using novelty-suppressed feeding (NSF) (Fig. 1n) and light–dark box (LDB) (Fig. 1o) tests. Interestingly, we did not find any mice that showed mechanical (Fig. 1m) or thermal (Fig. 1p) hyperalgesia in this experiment. To further determine whether the anxiety-like behavioral phenotype

was exclusive for the activation of nNOS-expressing neurons, we also microinjected AAV-hSyn-DO-hM3Dq-eGFP into the vmPFC of nNOS-Cre mice to stimulate Cre-negative neurons (Supplementary Fig. 4a, b) and found no alterations in anxiety-like behaviors (Supplementary Fig. 4c–e), but there was a clear trend suggesting an analgesic effect ($t_{24} = 2.040$, $p = 0.053$, Student's $t$-test).

**Fig. 1 Stimulation of vmPFC nNOS-expressing neurons induced anxiety-like behaviors. a–c** Withdrawal threshold of hind paw in response to von Frey hair stimulation (**a**), distance in center (**b**, left) and total distance (**b**, right) in an OF test, time in open arms (**c**, left) and total entries in arms (**c**, right) in an EPM test at day 3 after CFA (10 µl) or NS (10 µl) injection. $n = 14$ for NS and $n = 13$ for CFA. **d–f** Immunofluorescent images (**d**) and statistical data showing c-Fos$^+$ cells (**e**) and nNOS-expressing neurons co-labeled with c-Fos (**f**) in the vmPFC 4 h after CFA or NS injection. $n = 6$. **g** Scheme indicating the vmPFC area where AAV-hSyn-DIO-hM3Dq-eGFP (0.4 µl) and CNO (0.5 mM, 1 µl) or NS (1 µl) was microinjected into nNOS-Cre mice. **h** Coronal section showing the expression of hM3Dq-eGFP in the vmPFC of nNOS-Cre mice. Similar results were observed in 9 mice out of 10 mice. **i** Immunofluorescent images indicating the expression of hM3Dq-eGFP in vmPFC nNOS neurons. Arrows marked type I and arrowheads marked type II nNOS-expressing neurons. This experiment was conducted in triplicate with 5 mice and similar results were observed. **j** Bath application of CNO (5 µM) elicited action potential firing in neurons expressing hM3Dq-eGFP in brain slices of nNOS-Cre mice. **k–m** Distance in center (**k**, left) and total distance (**k**, right) in an OF test, time in open arms (**l**, left) and total entries in arms (**l**, right) in an EPM test, and withdrawal threshold of hind paw in a von Frey test (**m**) 21 days after AAV-hSyn-DIO-hM3Dq-eGFP microinjection. $n = 15$. **n–p** Latency to feed (**n**, left) and total feed (**n**, right) in NSF, time in side with light (**o**, left) and total distance (**o**, right) in LDB and withdrawal latency to thermal stimulation (**p**) 21 days after AAV-hSyn-DIO-hM3Dq-eGFP microinjection. $n = 14$. Data are the mean ± SEM; *$p < 0.05$, **$p < 0.01$, and ***$p < 0.001$ (unpaired two-tailed Student's t-test). Scale bars, 50 µm in (**d**) and (**i**), 1 mm in (**h**). Source data are provided as a Source data file. Exact p values and additional statistical information can be found in Source data.

Then, we delivered AAV-hSyn-DIO-hM4Di-eGFP into the vmPFC of nNOS-Cre mice to inhibit the activity of nNOS-expressing neurons and examined the chronic pain and anxiety-like behaviors induced by CFA injection (Fig. 2a). Chemogenetic inhibition of vmPFC nNOS-expressing neurons showed significant anxiolytic effects (OF: Fig. 2b; EPM: Fig. 2c). These anxiolytic effects did not come from increased locomotor activities (Fig. 2b, c). Since this chemogenetic inhibition did not produce a significant anxiolytic effect in normal mice (Supplementary Fig. 5), we determined that silencing vmPFC nNOS-expressing neurons rescued CFA-induced anxiety. Consistent with the normal withdrawal threshold observed when vmPFC nNOS-expressing neurons were stimulated, no analgesic effect was observed when this neuron population was silenced in the context of CFA-induced chronic pain (Fig. 2d). Furthermore, we induced chronic neuropathic pain in mice by spinal nerve ligation (SNL)[34,35] and investigated the influences of chemogenetic inhibition of vmPFC nNOS-expressing neurons (Fig. 2e). The results (Fig. 2f–h) were consistent with those from the CFA model. Together, the above experiments demonstrated that activation of nNOS-expressing neurons in the vmPFC is required for anxiety-like behaviors induced by chronic pain, though it was not responsible for hyperalgesia.

**The contribution of pPVT-vmPFC projections.** Subsequently, we explored the region innervating vmPFC nNOS-expressing neurons during chronic pain-induced anxiety. Since the projections from the PVT to the vmPFC are activated by neuropathic pain[36] and visceral pain[37], we examined c-Fos immunoreactivity in the PVT. As shown in Fig. 3a, almost twice as many c-Fos$^+$ cells per mm$^2$ were observed in the pPVT of CFA-injected mice compared with NS-injected mice, suggesting activation of the pPVT neurons during chronic pain. Most of these c-Fos$^+$ neurons were also CaMKIIα positive (Supplementary Fig. 6), indicating their glutamatergic properties.

To directly label the glutamatergic pPVT-vmPFC projections, we delivered AAV-CaMKIIα-hM3Dq-mCherry into the pPVT (Fig. 3b). The virus produced effective mCherry expression in cells at the injection site (Fig. 3c, left), and the mCherry$^+$ projects were observed in multiple layers of the vmPFC; there was more localization in layers 1 and 2/3 and less localization in layers 5 and 6 (Fig. 3c, middle and right). Efficient stimulation of hM3Dq-mCherry-expressing neurons was confirmed by increased AP firing after applying a CNO bath to brain slices (Fig. 3d). To obtain chemogenetic projection-specific activation of glutamatergic inputs from the pPVT to the vmPFC, CNO was microinjected into the vmPFC 21 days after AAV-CaMKIIα-hM3Dq-mCherry infection in the pPVT (Fig. 3b). The results indicated that chemogenetic activation of the glutamatergic pPVT-vmPFC

projections caused substantial mechanical (Fig. 3e) and thermal hyperalgesia (Fig. 3h) as well as anxiety-like behaviors (Fig. 3f, g, i, j). However, there were no impairments of locomotor activities (Fig. 3f, g, i, j). These effects mimicked CFA-induced chronic pain and anxiety-like behaviors.

To determine the role of pPVT-vmPFC projections in hyperalgesia and anxiety-like behaviors during chronic pain, we delivered AAV-CaMKIIα-hM4Di-mCherry into the PVT 18 days before CFA injection and microinjected CNO or NS into the vmPFC 3 days after the CFA injection (Fig. 3k). Efficient inhibition of hM4Dq-expressing neurons after CNO application was confirmed by the reduction in AP firing in brain slices (Fig. 3l). The results showed that chemogenetic inhibition of the projections from the pPVT to the vmPFC partially suppressed the CFA-induced decrease in the withdrawal threshold (Fig. 3m), indicating that there were analgesic effects. In parallel, this chemogenetic manipulation exhibited anxiolytic effects (OF: Fig. 3n; EPM: Fig. 3o). Chemogenetic manipulation had no significant effects on locomotor activities (Fig. 3n, o) and did not cause a marked anxiolytic effect in normal mice (Supplementary Fig. 7). Moreover, we repeated the experiment with another AAV, which expressed hM4Di (AAV-hSyn-HA-hM4Di-IRES-mCitrine) and obtained similar results (Supplementary Fig. 8). Additionally, we excluded the possibility that the behavioral changes were the nonspecific effects of CNO itself (Supplementary Fig. 9). Therefore, our results suggested that the pPVT-vmPFC projections were activated during chronic pain and contributed to both hyperalgesia and anxiety-like behaviors.

**pPVT-vmPFC projections target nNOS-expressing neurons.** Having established that the activation of nNOS-expressing neurons in the vmPFC and the excitation of pPVT-vmPFC projections are both required for anxiety-like behaviors induced by chronic pain, but only the latter is implicated in hyperalgesia, we hypothesized that nNOS-expressing neurons in the vmPFC are stimulated by pPVT inputs and then transform the signal from pain to anxiety. Indeed, compared with controls, we found many more nNOS$^+$ neurons in the vmPFC co-labeled with c-Fos (Fig. 4a) as well as increased total c-Fos$^+$ cells (Fig. 4b) with chemogenetic stimulation of pPVT-vmPFC projections. In contrast, the CFA-induced increase in the c-Fos$^+$ and nNOS$^+$/nNOS$^+$ ratio (Fig. 4d) and total c-Fos$^+$ cell number per mm$^2$ (Fig. 4e) in the vmPFC was suppressed when pPVT-vmPFC projections were inhibited by chemogenetics. Unlike the ratio of c-Fos$^+$ and nNOS$^+$/nNOS$^+$, the ratio of c-Fos$^+$ and nNOS$^+$/c-Fos$^+$ was not markedly changed (Fig. 4c, f).

Then, we directly determined whether the pPVT-vmPFC projections targeted nNOS-expressing neurons with photostimulation-induced synaptic responses. First, we microinjected an AAV encoding ChR2 (AAV-CaMKIIα-ChR2-eYFP) into the

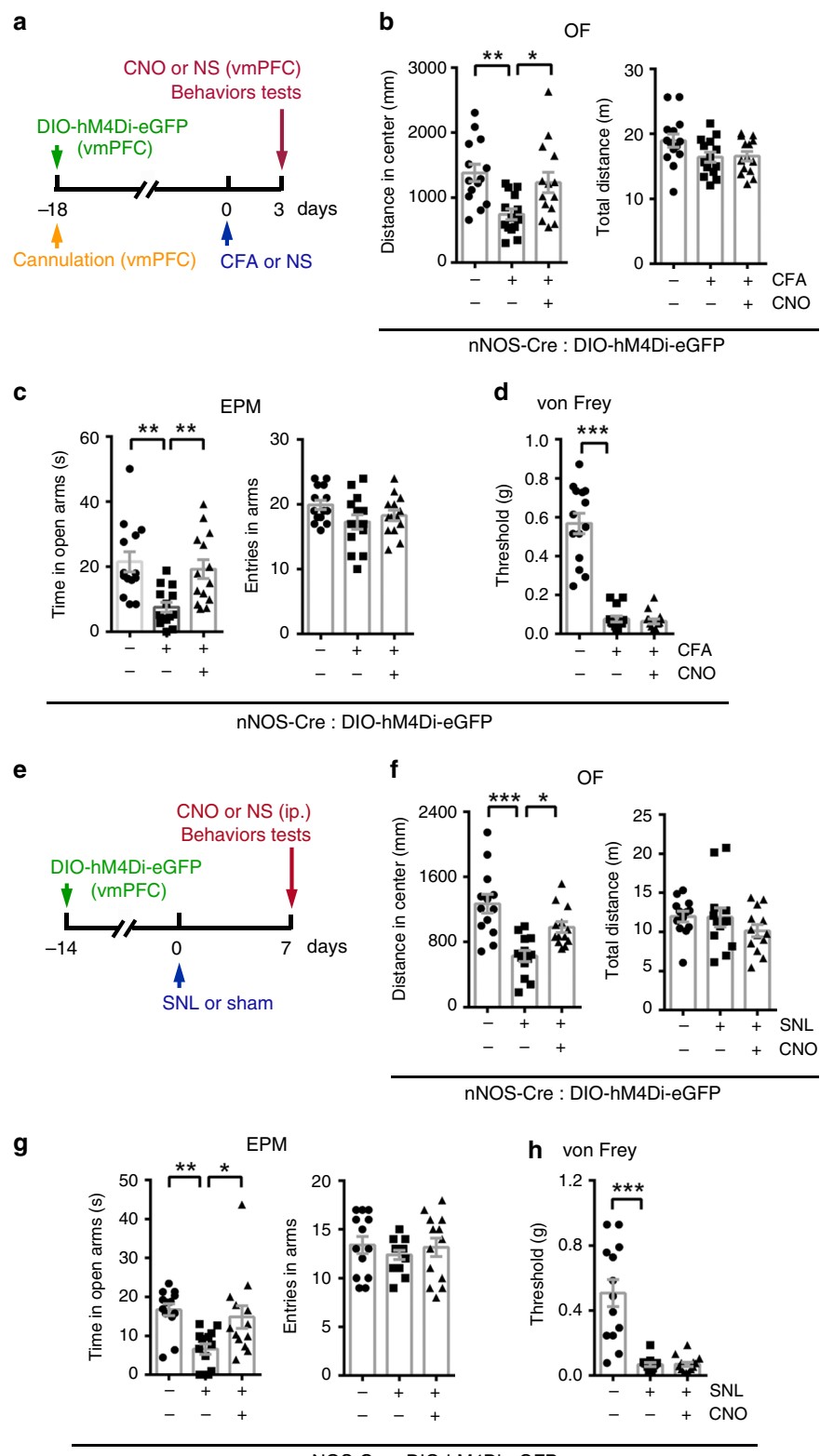

pPVT and confirmed the functionality of expressed ChR2 in brain slices by recording blue light-induced AP spikes in cells expressing ChR2-eYFP. Light stimulation induced strictly pulse-locked AP firing with high temporal fidelity (Fig. 4g). Subsequently, AAV-CaMKIIα-ChR2-eYFP and AAV-hSyn-DIO-mCherry were microinjected into the pPVT and vmPFC of nNOS-Cre mice, respectively (Fig. 4h). Three weeks later, mCherry[+] nNOS-expressing neurons in the vmPFC were

surrounded by ChR2-eYFP[+] projections from the pPVT (Fig. 4i, left). Then, nNOS-Cre mice were sacrificed for brain slice preparation. The vmPFC inputs from the pPVT were optically stimulated by blue light, and the evoked glutamatergic excitatory postsynaptic currents (eEPSCs) were recorded from a vmPFC nNOS-expressing neuron, which was visualized by Cre-dependent mCherry expression (Fig. 4i, right). As shown in Fig. 4j, eEPSCs were evoked on 41 out of 48 neurons by short

**Fig. 2 The role of vmPFC nNOS-expressing neurons in chronic pain and anxiety-like behaviors. a** Experimental design for (**b–d**). Ten microliters of CFA or NS was injected into the hind paw, and AAV-hSyn-DIO-hM4Di-eGFP (0.4 μl) and CNO (0.5 mM, 1 μl) or NS were microinjected into the vmPFC of nNOS-Cre mice on the indicated day. **b–d** Distance in center (**b**, left) and total distance (**b**, right) in an OF test, time in open arms (**c**, left) and total entries in arms (**c**, right) of an EPM test, and withdrawal threshold of hind paw in a von Frey test (**d**) 3 days after CFA or NS injection. $n = 14$. **e** Experimental design for (**f–h**). Chronic neuropathic pain was induced by spinal nerve ligation (SNL). AAV-hSyn-DIO-hM4Di-eGFP (0.4 μl) was microinjected into the vmPFC of nNOS-Cre mice, and CNO (2 mg per kg) or NS was injected intraperitoneally. **f–h** Distance in center (**f**, left) and total distance (**f**, right) in an OF test, time in open arms (**g**, left) and total entries in arms (**g**, right) of an EPM test and withdrawal threshold of the hind paw in a von Frey test (**h**) 7 days after SNL or sham operation. $n = 13$. Data are the mean ± SEM; *$p < 0.05$, **$p < 0.01$, and ***$p < 0.001$ (one-way ANOVA followed by Tukey's *post hoc* test). Source data are provided as a Source data file. Exact $p$ values and additional statistical information can be found in Source data.

photostimuli (5 ms duration) under voltage-clamp conditions; this result was demonstrated by their sensitivity to the combination of AMPA/kainate receptor antagonist 6-cyano-7-nitroquinoxaline-2,3-diones (CNQX) and NMDA receptor antagonist D-(−)-2-amino-5-phosphonopentanoic acid (AP-5). Synaptic responses were blocked by tetrodotoxin (TTX) (Fig. 4k), confirming that the responses were due to AP firing in presynaptic fibers arising from pPVT. The application of 4-aminopyridine (4-AP), a blocker of voltage-gated K$^+$ channels, rescued TTX-blocked eEPSCs (Fig. 4k), indicating that the recorded eEPSCs were monosynaptic.

Furthermore, we determined whether inhibition of nNOS-expressing neurons in the vmPFC could attenuate anxiety-like behaviors following pPVT-vmPFC projection stimulation. AAV-CaMKIIα-hM3Dq-mCherry and AAV-hSyn-DIO-hM4Di-eGFP (or AAV-hSyn-DIO-eGFP) were microinjected into the pPVT and vmPFC of nNOS-Cre mice, respectively (Fig. 4l). Three weeks later, the nNOS-Cre mice were intraperitoneally injected with CNO to simultaneously stimulate pPVT-vmPFC projections and inhibit nNOS-expressing neurons in the vmPFC. Given that anxiety-like behaviors were induced by chemogenetic stimulation of pPVT-vmPFC projections (Fig. 3e–g), when nNOS-expressing neurons in the vmPFC were inhibited, there was an increased distance in the center of the OF (Fig. 4m) and increased time in the open arms of the EPM (Fig. 4n); there was no change in locomotor activities (Fig. 4m, n), suggesting anxiolytic effects. Moreover, mechanical hyperalgesia following chemogenetic stimulation of pPVT-vmPFC projections was not influenced by chemogenetic inhibition of nNOS-expressing neurons in the vmPFC (Fig. 4o). Taken together, the findings indicated that chronic pain stimulated glutamatergic pPVT-vmPFC projections that targeted nNOS-expressing neurons, and excitation of nNOS-expressing neurons in the vmPFC transformed pain signals into anxiety signals, leading to anxiety-like behaviors.

**Enhanced AMPAR trafficking in vmPFC**. To identify the downstream signals of nNOS-expressing neuron activation and NO production, we tested AMPARs; AMPAR trafficking to the plasma membrane is regulated by nitrosylation[38–40], and they function in the modulation of pain and anxiety[12,21]. As expected, vmPFC microinjection of the AMPA/kainate receptor antagonist CNQX 3 days after CFA injection did not reduce chronic pain (Supplementary Fig. 10a), but it completely abolished chronic pain-induced anxiety-like behaviors, as shown by both OF (Supplementary Fig. 10b) and EPM (Supplementary Fig. 10c) tests. Additionally, the locomotor activities of mice showed no marked changes (Supplementary Fig. 10b, c). To further confirm the AMPAR function in chronic pain-induced anxiety, AMPAR-dependent spontaneous excitatory postsynaptic currents (sEPSCs) and the AMPA/NMDA ratio of layer 2/3 pyramidal neurons in the vmPFC were recorded with brain slices prepared from CFA- and NS-injected mice. AAV-CaMKIIα-mCherry was microinjected into the vmPFC to identify pyramidal neurons 21 days before recording. The amplitude of sEPSCs in the CFA

group was substantially increased compared to that of the NS group (Fig. 5a, b), and the frequency of sEPSCs in the two groups was not significantly different (Fig. 5a, c). The AMPA/NMDA ratio was also obviously increased in the CFA group (Fig. 5d, e). Moreover, the GluA1 and GluA2 subunits on the plasma membrane surface were dramatically augmented 3 days after CFA injection (Fig. 5f–h), which was consistent with the increased sEPSC amplitude and AMPA/NMDA ratio. These results indicated that chronic pain enhanced AMPAR-dependent excitatory synaptic transmission and AMPAR trafficking in the vmPFC.

Given that the activation of nNOS-expressing neurons in the vmPFC was required for CFA-induced anxiety-like behaviors, we next examined whether AMPAR trafficking and function were controlled by this neuron population. AAV-CaMKIIα-mCherry and AAV-hSyn-DIO-hM3Dq-eGFP were microinjected into the vmPFC of nNOS-Cre mice for identification of pyramidal neurons and chemogenetic stimulation of nNOS neurons in the vmPFC, respectively (Fig. 5i). Three weeks later, the nNOS-Cre mice were sacrificed for brain slice preparation. AMPAR-dependent sEPSCs and the AMPA/NMDA ratio of the mCherry$^+$/eGFP$^-$ neurons (Fig. 5j) in layers 2/3 were recorded after 30 min of CNO or NS bath application. The mCherry$^+$/eGFP$^+$ neurons were excluded to avoid the direct activation of recorded neurons by CNO. In agreement with the results from the CFA model, chemogenetic stimulation of nNOS-expressing neurons by CNO was found to increase the amplitude of pyramidal neurons' sEPSCs without obvious influence on the frequency (Fig. 5k–m). The AMPA/NMDA ratio was also obviously increased in the CNO group (Fig. 5n, o). More importantly, CNO could not increase the sEPSC amplitude (Fig. 5k–m) or AMPA/NMDA ratio (Fig. 5n, o) when it was applied in combination with 2-(4-carboxyphenyl)-4,4,5,5-tetra-methylimidazoline-1-oxyl-3-oxide (C-PTIO), which is an NO scavenger; this result indicates that the augmentation of vmPFC AMPAR function after activation of nNOS-expressing neurons was dependent on NO production. Additionally, we measured the surface expression of the GluA1 and GluA2 subunits in the vmPFC of AAV-hSyn-DIO-hM3Dq-eGFP-infected nNOS-Cre mice 30 min after CNO or NS injection, and we found AMPAR trafficking to the plasma membrane was enhanced in CNO-injected mice compared with that of NS-injected mice (Fig. 5p–r). The results indicated that stimulation of nNOS-expressing neurons in the vmPFC promotes AMPAR trafficking through NO production, resulting in enhanced AMPAR function.

Furthermore, we investigated whether nNOS-expressing neuron activity was required for AMPAR trafficking and enhanced glutamatergic plasticity during chronic pain induced by CFA. AAV-CaMKIIα-mCherry and AAV-hSyn-DIO-hM4Di-eGFP were microinjected into the vmPFC of nNOS-Cre mice for identification of pyramidal neurons and chemogenetic inhibition of nNOS neurons in the vmPFC, respectively. Three weeks later, the nNOS-Cre mice were sacrificed to enable the collection of electrophysiological recordings of mCherry$^+$/eGFP$^-$ neurons in

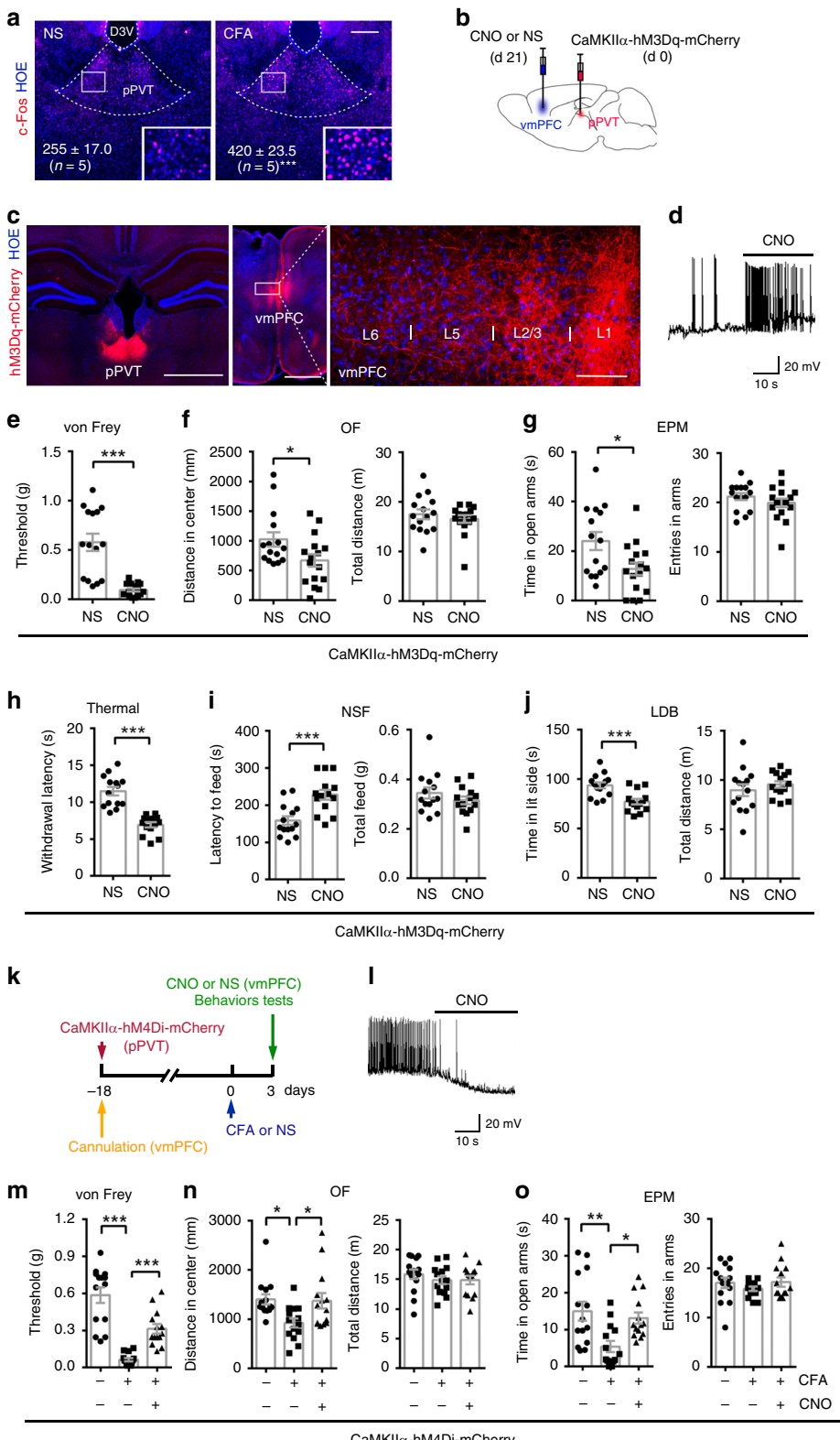

layer 2/3 of the vmPFC after 30 min of bathing with CNO or NS (Fig. 6a). AMPAR-dependent miniature excitatory postsynaptic currents (mEPSCs) were recorded instead of sEPSCs to exclude the influence of network activity, and the AMPA/NMDA ratio was also recorded. The results indicated that chemogenetic inhibition of nNOS-expressing neurons by CNO significantly suppressed the CFA-induced increase in mEPSC amplitude

(Fig. 6b–d) and AMPA/NMDA ratio (Fig. 6e, f) of pyramidal neurons, though it did not have obvious effects on brain slices from NS-injected mice.

Taken together, these electrophysiological experiments demonstrated that the enhancement of excitatory synaptic transmission in the vmPFC that was mediated by AMPAR trafficking was responsible for chronic pain-induced anxiety.

**Fig. 3 The role of glutamatergic pPVT-vmPFC projections in CFA-induced chronic pain and anxiety-like behaviors. a** Immunofluorescent images and statistical data showing the c-Fos+ cells in pPVT 4 h after CFA (10 µl) or NS injection. **b** Scheme indicating the pPVT area where AAV-CaMKIIα-hM3Dq-mCherry (0.2 µl) was microinjected and the vmPFC area where CNO (0.5 mM, 1 µl) or NS (1 µl) was microinjected. **c** Coronal sections illustrating hM3Dq-mCherry-positive pPVT neurons at the injection site (**c**, left) and their axon terminals projected to the vmPFC (**c**, middle). A magnified image (**c**, right) shows the asterisk-marked area. Similar results were observed in 6 mice out of 6 mice. **d** Bath application of CNO (5 µM) elicited action potential firing in neurons expressing hM3Dq-eGFP in brain slices. **e–g** Withdrawal threshold of hind paw in a von Frey test (**e**), distance in center (**f**, left) and total distance (**f**, right) of an OF test, and time in open arms (**g**, left) and total entries in arms (**g**, right) of an EPM test 21 days after AAV-CaMKIIα-hM3Dq-mCherry microinjection. $n = 15$ for NS and $n = 16$ for CNO. **h–j** Withdrawal latency to thermal stimulation (**h**), latency to feed (**i**, left) and total feed (**i**, right) during NSF, time in the side with light (**j**, left) and total distance (**j**, right) in a LDB for nNOS-Cre mice 21 days after AAV-CaMKIIα-hM3Dq-mCherry microinjection. $n = 14$. **k** Experimental design for (**l–o**). CFA (10 µl) or NS was injected into the hind paw, AAV-CaMKIIα-hM4Di-mCherry (0.2 µl) and CNO (0.5 mM, 1 µl) or NS (1 µl) were microinjected into the PVT and vmPFC, respectively, on the indicated day. **l** Bath application of CNO (5 µM) abolished action potential firing in neurons expressing hM4Di-mCherry in brain slices. **m–o** Withdrawal threshold of hind paw was measured with a von Frey test (**m**), distance in center (**n**, left) and total distance (**n**, right) in an OF test, and time in open arms (**o**, left) and total entries in arms (**o**, right) in an EPM test 3 days after CFA or NS injection. $n = 14$. Data are the mean ± SEM; *$p < 0.05$, **$p < 0.01$, and ***$p < 0.001$ (unpaired two-tailed Student's $t$-test for (**e–j**); one-way ANOVA followed by Tukey's *post hoc* test for (**m–o**)). Scale bars, 200 µm in (**a**), 1 mm in (**c**, left) and (**c**, middle), and 50 µm in (**c**, right). Source data are provided as a Source data file. Exact $p$ values and additional statistical information can be found in Source data.

**AMPAR function and classic neurotransmitter release**. Next, we examined the contribution of classic neurotransmitter release from nNOS-expressing neurons to the modulation of AMPAR function in the vmPFC during chronic pain-induced anxiety. nNOS-expressing neurons are traditionally regarded as GABAergic interneurons[29], but it was recently reported that they were glutamatergic in some brain regions[41]. We microinjected AAV-EF1a-DIO-ChR2-eYFP and AAV-CaMKIIα-mCherry into the vmPFC of nNOS-Cre mice (Supplementary Fig. 11a), and brain slices were prepared 3 weeks later. nNOS-expressing neurons were optically stimulated by blue light, and eEPSC or eIPSC data from the mCherry+/eYFP− layer 2/3 pyramidal neurons of the vmPFC were collected. mCherry+/eYFP+ neurons were excluded to avoid the direct effects of blue light on the analyzed neurons. By short photostimuli (5 ms duration) under voltage-clamp conditions, either EPSCs or IPSCs could be evoked (Supplementary Fig. 11b), indicating that layer 2/3 pyramidal neurons in the vmPFC are regulated by local nNOS-expressing neurons through both GABAergic and glutamatergic synapses. These recordings were performed with bath application of TTX and 4-AP, guaranteeing monosynaptic transmission.

Vesicular GABA transporters (VGATs) and vesicular glutamate transporters (VGLUTs) are pivotal for glutamate[42] and GABA[43] release from neurons, respectively. To specifically inhibit glutamate and GABA release from nNOS-expressing neurons, we used AAV-CMV-flex-shVGLUT-eGFP and AAV-CMV-flex-shVGAT-eGFP to enable shRNA-mediated conditional knock-down of VGLUT (Fig. 7a) and VGAT (Fig. 7c), respectively, in nNOS-Cre mice. When nNOS-expressing neurons were stimulated by optogenetics, light-evoked monosynaptic EPSCs on vmPFC layer 2/3 pyramidal neurons (eYFP−) were much more rare in the shVGLUT group (63.3%, 19 out 43 neurons) than they were in the shControl group (86.8%, 33 out 38 neurons), and the eEPSC amplitude was also significantly decreased by shVGLUT (Fig. 7b), confirming the effective VGLUT knockdown and reduced glutamate release from nNOS-expressing neurons. Alternatively, though light-evoked monosynaptic IPSCs were slightly decreased in the shVGAT group (83.7%, 36 out 43 neurons) compared with that of the shControl group (89.5%, 34 out 40 neurons), the eIPSC amplitude was significantly decreased by shVGAT (Fig. 7d), confirming the effective VGAT knockdown and reduced GABA release from nNOS-expressing neurons. Then, we determined whether shVGLUT or shVGAT prevented the augmentation of AMPAR function during chronic pain-induced anxiety by collecting AMPAR-dependent sEPSC recordings from brain slices from CFA model mice (Fig. 7e); we found that the CFA-induced amplitude increase of sEPSCs in layer 2/3 pyramidal neurons of the vmPFC was not suppressed by specific inhibition of glutamate or GABA release in nNOS-expressing neurons (Fig. 7f, g). Therefore, the results revealed that the modulation of AMPAR function in the vmPFC by nNOS-expressing neuron activation during chronic pain-induced anxiety is independent of classic neurotransmitter release. Decreased AMPAR frequency in the shVGLUT group (Fig. 7h) might result from reduced presynaptic glutamate release.

**The role of diffusing NO**. Finally, we wanted to determine how nNOS-expressing neuron activation promoted vmPFC AMPAR trafficking and resulted in anxiety-like behaviors during chronic pain. We assessed nNOS protein levels and enzyme activity in the vmPFC at day 3 after CFA injection when chronic pain-induced anxiety-like behaviors were obvious. The nNOS protein level was not changed in CFA-injected mice (Fig. 8a), but the enzyme activity was significantly increased (Fig. 8b) compared with that of NS-injected mice. We also found increased nNOS enzyme activity when vmPFC nNOS-expressing neurons were chemogenetically activated (Fig. 8c). Thus, there is a possibility that nNOS-expressing neurons in the vmPFC mediate chronic pain-induced anxiety-like behaviors through increased nNOS enzyme activity and NO production.

To assess this possibility, we employed the nNOS-specific inhibitor N5-(1-imino-3-butenyl)-L-ornithine (L-VNIO) and the NO donor 3,3-bis(aminoethyl)-1-hydroxy-2-oxo-1-triazene (DETA/NONOate)[44]. Although local administration of L-VNIO in the vmPFC did not produce a significant anxiolytic effect in normal mice (Supplementary Fig. 12), it actually reversed the chronic inflammatory pain-induced anxiety-like behaviors at day 3 after CFA injection (Fig. 8d); additionally, there was no influence on locomotor activities (Fig. 8d, e) and mechanical hyperalgesia (Fig. 8f). Furthermore, we induced chronic neuropathic pain by SNL and found that inhibition of nNOS enzyme activity had similar anxiolytic effects and did not affect locomotor activities (Fig. 8g, h) or mechanical hyperalgesia (Fig. 8i) at day 7 after surgery, indicating the necessary role of the nNOS enzyme in different chronic pain-induced anxiety-like behaviors. In contrast, microinjection of DETA/NONOate in the vmPFC led to anxiety-like behaviors without influencing locomotor activities (Supplementary Fig. 13a, b) and algesia (Supplementary Fig. 13c). The results from L-VNIO and DETA/NONOate suggested that nNOS enzyme activity and NO production were essential for chronic pain-induced anxiety. To directly investigate the role of the nNOS–NO pathway in mediating anxiety-like behaviors induced by the activation of nNOS-expressing neurons, we took nNOS-Cre mice that had been microinjected with

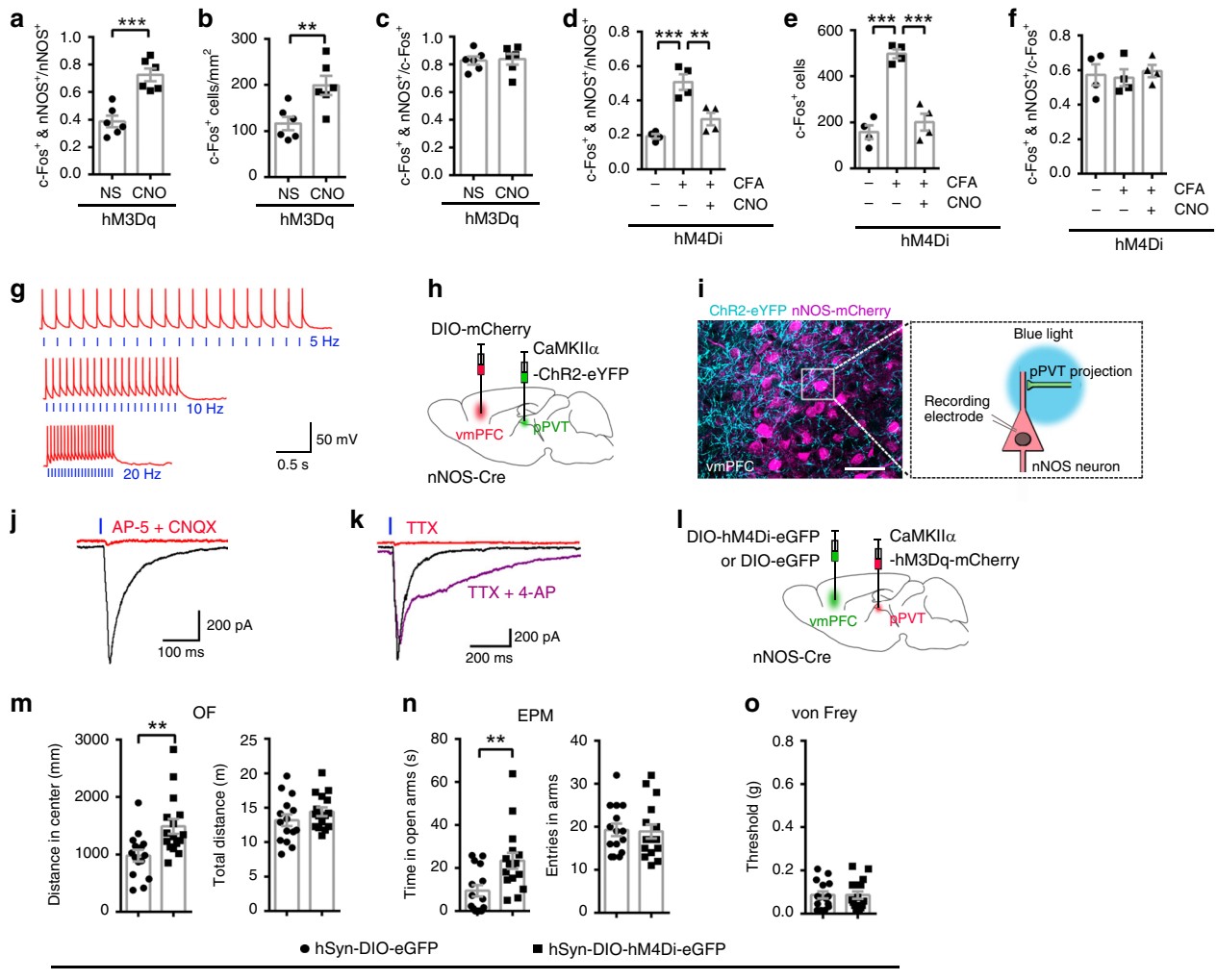

**Fig. 4 The projections from pPVT targeted nNOS-expressing neurons in the vmPFC. a–c** Dot plots showing the ratio of c-Fos and nNOS double-positive neurons to total nNOS-positive neurons (**a**), the number of c-Fos+ cells (**b**), and the ratio of c-Fos and nNOS double-positive neurons to total c-Fos-positive neurons (**c**) in the vmPFC when pPVT-vmPFC projections were excited by chemogenetics. $n = 6$. **d–f** Dot plots showing the ratio of c-Fos and nNOS double-positive neurons to total nNOS-positive neurons (**d**), the number of c-Fos+ cells (**e**), and the ratio of c-Fos and nNOS double-positive neurons to total c-Fos-positive neurons (**f**) in the vmPFC of CFA-injected mice when pPVT-vmPFC projections were inhibited by chemogenetics. $n = 4$. **g** Action potentials were evoked in a ChR2-eYFP-expressing neuron in pPVT in current-clamp mode by photostimuli pulses (blue light: 465 nm, 43.7 mW, 5 ms duration) at various frequencies (blue vertical lines). **h** Scheme indicating the pPVT area where AAV-CaMKIIα-ChR2-eYFP (0.2 µl) was microinjected and the vmPFC area where AAV-hSy-DIO-mCherry (0.4 µl) was microinjected. **i** Experimental setup for recording photostimulation-induced synaptic responses. ChR2-eYFP+ projections from the pPVT were excited by blue light (465 nm, 43.7 mW, 5 ms duration), and evoked EPSCs were recorded on mCherry+ nNOS-expressing neurons in the vmPFC. **j**, **k** Traces of light-evoked EPSCs recorded on nNOS-expressing neurons in the vmPFC, blocked by CNQX (10 µM) and AP-5 (50 µM) (**j**), or blocked by TTX (0.5 µM) and rescued by 4-AP (100 µM) (**k**). **l** Scheme indicating the pPVT area where AAV-CaMKIIα-hM3Dq-mCherry (0.2 µl) was microinjected, and the vmPFC area where AAV-hSyn-DIO-hM4Di-eGFP or AAV-hSyn-DIO-eGFP (0.4 µl) was microinjected into nNOS-Cre mice. **m–o** Distance in center (**m**, left) and total distance (**m**, right) in an OF test, time in open arms (**n**, left) and total entries in arms (**n**, right) in an EPM test, and withdrawal threshold of hind paw in a von Frey test (**o**) 30 min after CNO intraperitoneal injection (2 mg per kg) at day 21 after AAV microinjections. $n = 15$ and 16, respectively. Data are the mean ± SEM; **$p < 0.01$, and ***$p < 0.001$ (unpaired two-tailed Student's t-test for (**a–c**) and (**m–o**); one-way ANOVA followed by Tukey's post hoc test for (**d–f**)). Scale bar, 50 µm in (**i**). Source data are provided as a Source data file. Exact $p$ values and additional statistical information can be found in Source data.

AAV-hSyn-DIO-hM3Dq-eGFP in the vmPFC, and we treated them with L-VNIO or NS 30 min after CNO delivery (day 21). As expected, anxiolytic effects of L-VNIO in OF (Fig. 8j) and EPM tests were observed (Fig. 8k), which was in agreement with the results from the CFA model. Meanwhile, the locomotor activities of mice did not change significantly (Fig. 8j, k). Taken together, the results suggested that signal transduction shift from chronic pain to anxiety in the vmPFC after activation of nNOS-expressing neurons was dependent on the nNOS–NO pathway.

NO regulates AMPARs by targeting the AMPAR-interacting proteins stargazin and N-ethylmaleimide sensitive factor (NeSF). These two proteins are principal determinants of AMPAR surface expression. S-nitrosylation of stargazin and NeSF elicits enhanced surface expression of GluA1 and GluA2, respectively[38,39]. Therefore, we examined S-nitrosylated stargazin and NeSF in the vmPFC. A biotin-switch assay showed that the S-nitrosylation level of stargazin in the vmPFC of CFA-injected mice was increased by approximately 3-fold compared to that of

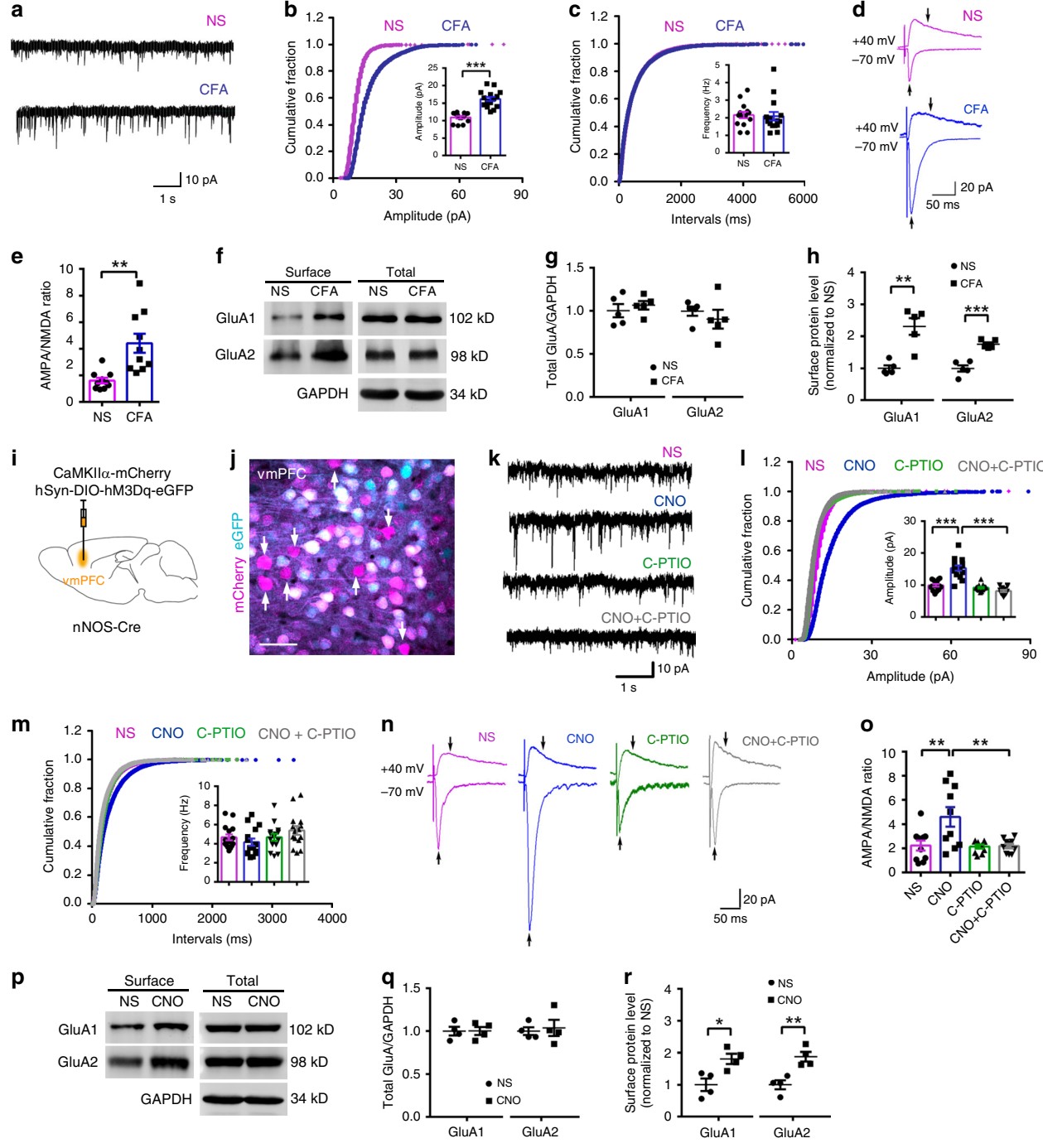

NS-injected mice (Supplementary Fig. 14a–c) at day 3 after injection, which is when chronic pain-induced anxiety was obvious. The level of S-nitrosylated NeSF also increased significantly (Supplementary Fig. 14a–c), though the change was not as dramatic as that of S-nitrosylated stargazin. Furthermore, we measured the S-nitrosylation levels of these two proteins in AAV-hSyn-DIO-hM4Di-eGFP-infected nNOS-Cre mice at day 3 after CFA was injected to induce anxiety. As expected, when vmPFC nNOS-expressing neurons were silenced by CNO, the S-nitrosylation of stargazin and NeSF induced by CFA was suppressed (Supplementary Fig. 14d–f), which was consistent with the alleviation of anxiety induced by chronic pain. Therefore, the results indicated that S-nitrosylation of stargazin and NeSF in the vmPFC by diffusing NO occurs following the

activation of nNOS-expressing neurons, which might explain the mechanism of AMPAR trafficking to the plasma membrane.

## Discussion

The results presented here demonstrate the crucial role of vmPFC nNOS-expressing neurons in chronic pain-induced anxiety. Pain signal transformation into an anxiety signal occurs through stimulation of this neuron population by glutamatergic pPVT-vmPFC inputs; then, S-nitrosylation of proteins interacting with AMPARs occurs via diffusing NO, which is generated by activated nNOS enzyme to promote AMPAR trafficking to the plasma membrane (summarized in Fig. 9). Consequently, augmented vmPFC AMPAR functions mediate chronic pain-induced anxiety. This is a new insight into the neural circuit and the

**Fig. 5 Enhanced AMPAR trafficking and function in the vmPFC were associated with chronic pain-induced anxiety. a** Representative traces of sEPSCs in layer 2/3 pyramidal neurons of the vmPFC 3 days after CFA or NS injection. **b**, **c** Cumulative fraction plots of sEPSC amplitude (**b**, mean amplitude in inset) and interevent intervals (**c**, mean frequency in inset). $n = 14$ (from 5 mice) and 15 (from 6 mice) neurons for the NS and CFA groups, respectively. **d** Representative traces of evoked EPSCs (eEPSCs) in the presence of BMI in layer 2/3 pyramidal neurons of the vmPFC 3 days after CFA or NS injection. Arrows indicate that AMPAR-EPSCs were measured at the peak at a holding potential of −70 mV, and NMDAR-EPSCs were measured 50 ms after stimulation at a holding potential of +40 mV. **e** Dot plot showing the AMPA/NMDA ratio from (**d**). $n = 10$ neurons (from 4 mice). (**f**–**h**) Representative western blots (**f**) and dot plots showing total (**g**) and surface (**h**) protein levels of GluA1 and GluA2 at day 3 after CFA or NS injection. **i** Scheme indicating the vmPFC area where AAV-hSyn-DIO-hM3Dq-eGFP (0.2 µl) and AAV-CaMKIIα-mCherry (0.2 µl) were microinjected into nNOS-Cre mice. $n = 5$. **j** Immunofluorescent image of vmPFC showing mCherry⁺/eGFP⁻ neurons (arrows indicated), form which sEPSCs and eEPSCs were recorded. This experiment was conducted in triplicate with 5 mice and similar results were observed. **k** Representative traces of sEPSCs in layer 2/3 pyramidal neurons of the vmPFC 3 weeks after AAV injection into nNOS-Cre mice. **l**, **m** Cumulative fraction plots of sEPSC amplitude (**l**, mean amplitude in inset) and interevent intervals (**m**, mean frequency in inset). $n = 15$ neurons (from 6 mice). **n** Representative traces of eEPSCs in the presence of BMI in layer 2/3 pyramidal neurons of the vmPFC 3 weeks after AAV injection into nNOS-Cre mice. Arrows indicate that AMPAR-EPSCs were measured at their peak at a holding potential of −70 mV, and NMDAR-EPSCs were measured 50 ms after stimulation at a holding potential of +40 mV. **o** Bar graph showing the AMPA/NMDA ratio. $n = 10$ neurons (from 4 mice). **p**–**r** Representative western blots (**p**) and dot plots showing total (**q**) and surface (**r**) protein levels of GluA1 and GluA2 30 min after CNO (2 mg per kg, i.p.) or NS treatment at day 21 after AAV microinjection. $n = 4$. Data are the mean ± SEM; *$p < 0.05$, **$p < 0.01$, and ***$p < 0.001$ (unpaired two-tailed Student's $t$-test for (**b**), (**c**), (**e**), (**g**), (**h**), (**q**), and (**r**); one-way ANOVA followed by Tukey's *post hoc* test for (**i**), (**m**), and (**o**)). Scale bar, 50 µM in (**j**). Source data are provided as a Source data file. Exact $p$ values and additional statistical information can be found in Source data.

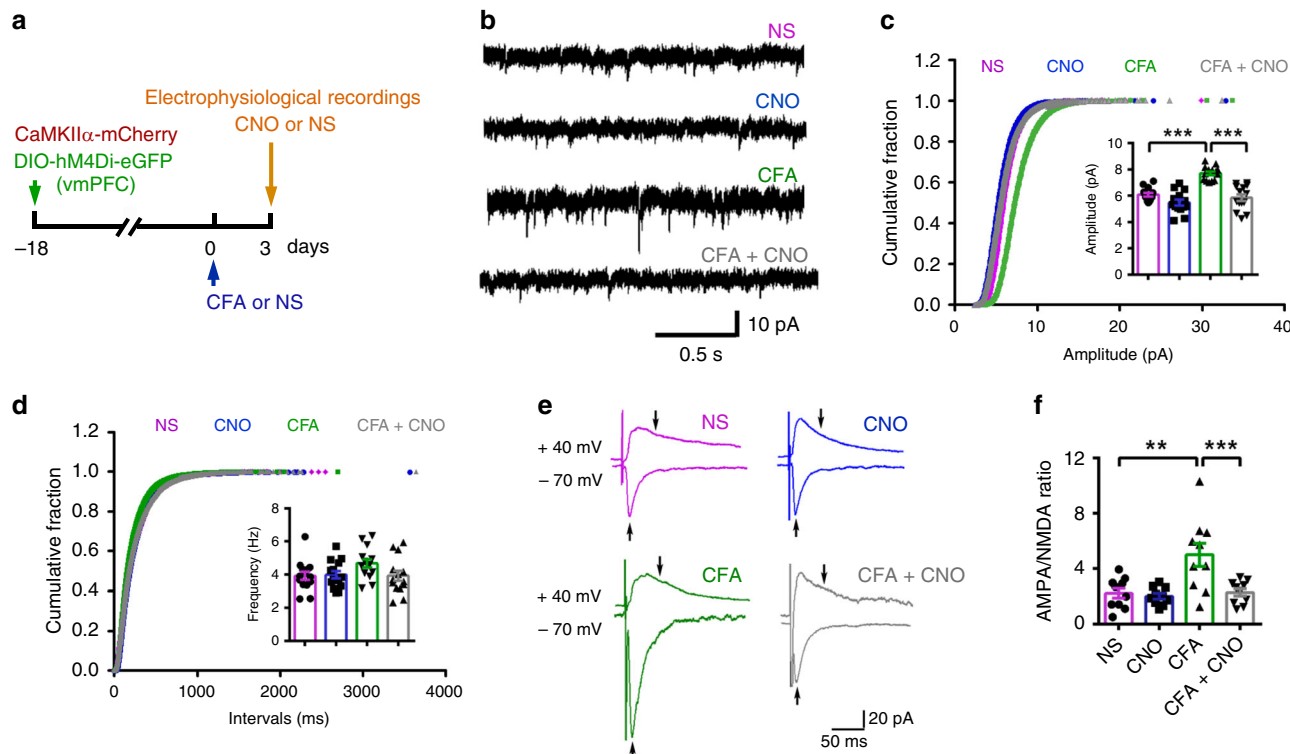

**Fig. 6 Chemogenetic inhibition of nNOS-expressing neurons suppressed CFA-induced AMPAR trafficking in the vmPFC. a** Experimental design for (**b**–**f**). Ten microliters of CFA or NS was injected into the hind paw, and AAV-hSyn-DIO-hM4Di-eGFP (0.2 µl) and AAV-CaMKIIα-mCherry (0.2 µl) were microinjected into the vmPFC of nNOS-Cre mice on the indicated day. **b** Representative traces of mEPSCs in layer 2/3 pyramidal neurons of the vmPFC 3 weeks after AAV injection into nNOS-Cre mice. **c**, **d** Cumulative fraction plots of mEPSC amplitude (**c**, mean amplitude in inset) and interevent intervals (**d**, mean frequency in inset). $n = 14$ neurons (from 6 mice). **e** Representative traces of eEPSCs in the presence of BMI in layer 2/3 pyramidal neurons of the vmPFC 3 weeks after AAV injection into nNOS-Cre mice. Arrows indicate that AMPAR-EPSCs were measured at their peak at a holding potential of −70 mV, and NMDAR-EPSCs were measured 50 ms after stimulation at a holding potential of +40 mV. **f** Dot plot showing the AMPA/NMDA ratio from (**e**). $n = 10$ neurons (from 4 mice). Data are the mean ± SEM; **$p < 0.01$, and ***$p < 0.001$. (one-way ANOVA followed by Tukey's *post hoc* test). Source data are provided as a Source data file. Exact $p$ values and additional statistical information can be found in Source data.

molecular mechanisms of anxiety induced by chronic pain, especially the finding that nNOS-expressing neurons in the vmPFC are stimulated by a pain signal from the pPVT and then transform the pain signal into the anxiety signal. Previous studies have mostly investigated pain and anxiety independently[7]. Until recently, few reports focused on the interaction of chronic pain and anxiety[13,26], and they indicated the common pathophysiology of both chronic pain and anxiety. Our results suggest that there are both similarities and differences in chronic pain and anxiety.

In this study, we demonstrated that augmentation of glutamatergic transmission through enhanced AMPAR trafficking in

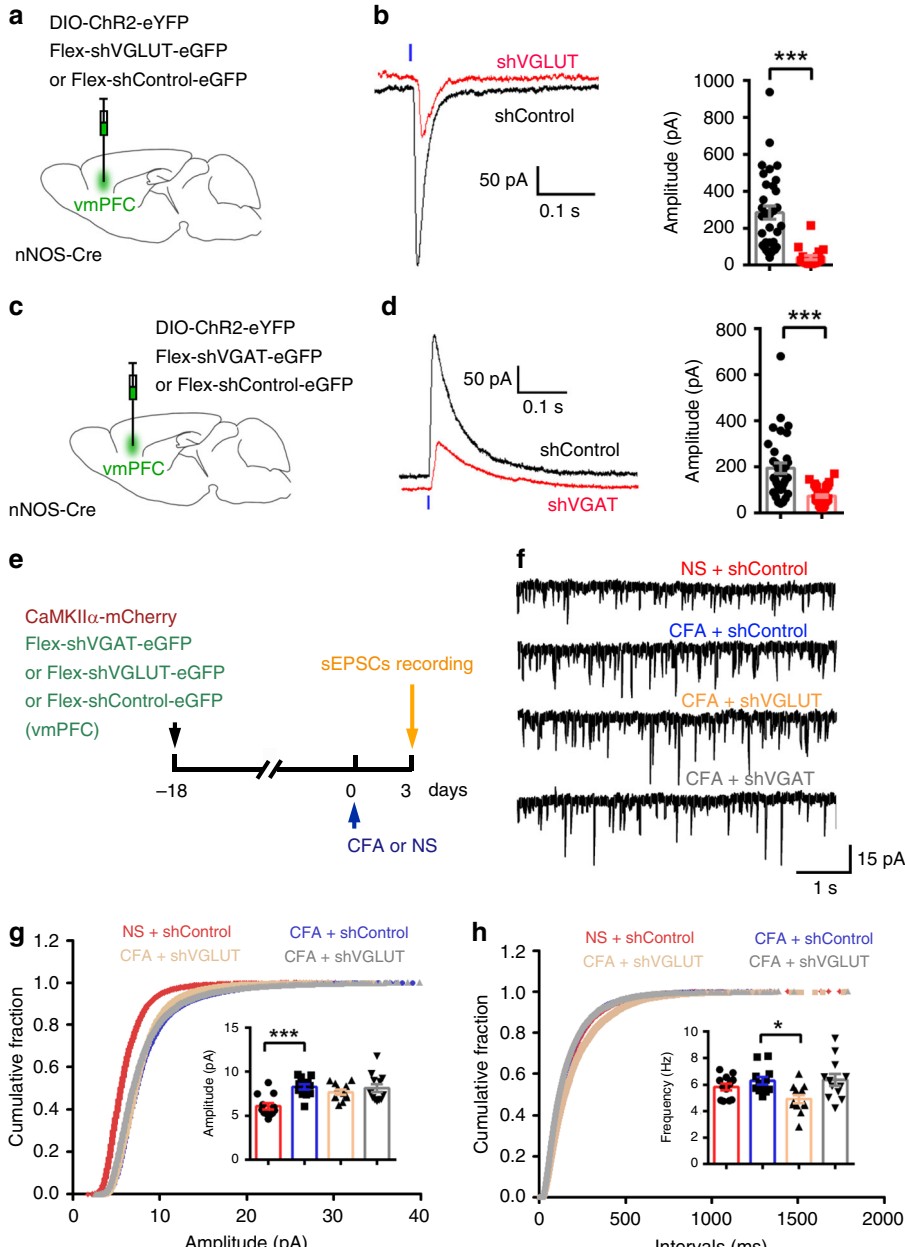

**Fig. 7 Inhibiting glutamate or GABA release from nNOS-expressing neurons did not affect AMPAR-dependent sEPSCs in the vmPFC. a** Scheme indicating the vmPFC area where AAV-EF1a-DIO-ChR2-eYFP (0.2 μl) and AAV-CMV-flex-shVGLUT-eGFP or AAV-CMV-flex-shControl-eGFP (0.2 μl) were microinjected into nNOS-Cre mice. **b** Traces of light-evoked EPSCs (left) and dot plot of eEPSC amplitude (right) recorded on layer 2/3 pyramidal neurons in the vmPFC 3 weeks after the mixed AAVs were microinjected. $n = 33$ or 19 neurons (from 5 mice). **c** Scheme indicating the vmPFC area where AAV-EF1a-DIO-ChR2-eYFP (0.2 μl) and AAV-CMV-flex-shVGAT-eGFP or AAV-CMV-flex-shControl-eGFP (0.2 μl) were microinjected into nNOS-Cre mice. **d** Traces of light-evoked IPSCs (left) and dot plot of eIPSC amplitude (right) recorded on layer 2/3 pyramidal neurons in the vmPFC 3 weeks after the mixed AAVs were microinjected. $n = 34$ or 36 neurons (from 6 mice). eYFP[+] nNOS-expressing neurons were excited by blue light (465 nm, 43.7 mW, 5 ms duration), and eYFP[−] pyramidal neurons in layer 2/3 of the vmPFC were assessed in the presence of TTX (0.5 μM) and 4-AP (100 μM). **e** Experimental design for (**f–h**). Ten microliters of CFA or NS was injected into the hind paw, and a mixture of AAV-CaMKIIα-mCherry (0.2 μl) and AAV-CMV-flex-shVGLUT-eGFP or AAV-CMV-flex-shVGAT-eGFP or AAV-CMV-flex-shControl-eGFP (0.2 μl) was microinjected into the vmPFC of nNOS-Cre mice on the indicated day. **f** Representative traces of sEPSC in layer 2/3 pyramidal neurons of the vmPFC. The pyramidal neurons were identified by mCherry expression. **g**, **h** Cumulative fraction plots of sEPSC amplitude (**g**, mean amplitude in inset) and interevent intervals (**h**, mean frequency in inset). $n = 12$ neurons (from 5 mice) for each group. Data are the mean ± SEM; *$p < 0.05$, **$p < 0.01$, and ***$p < 0.001$ (unpaired two-tailed Student's $t$-test for (**b**) and (**d**); one-way ANOVA followed by Tukey's *post hoc* test for (**g**) and (**h**)). Source data are provided as a Source data file. Exact $p$ values and additional statistical information can be found in Source data.

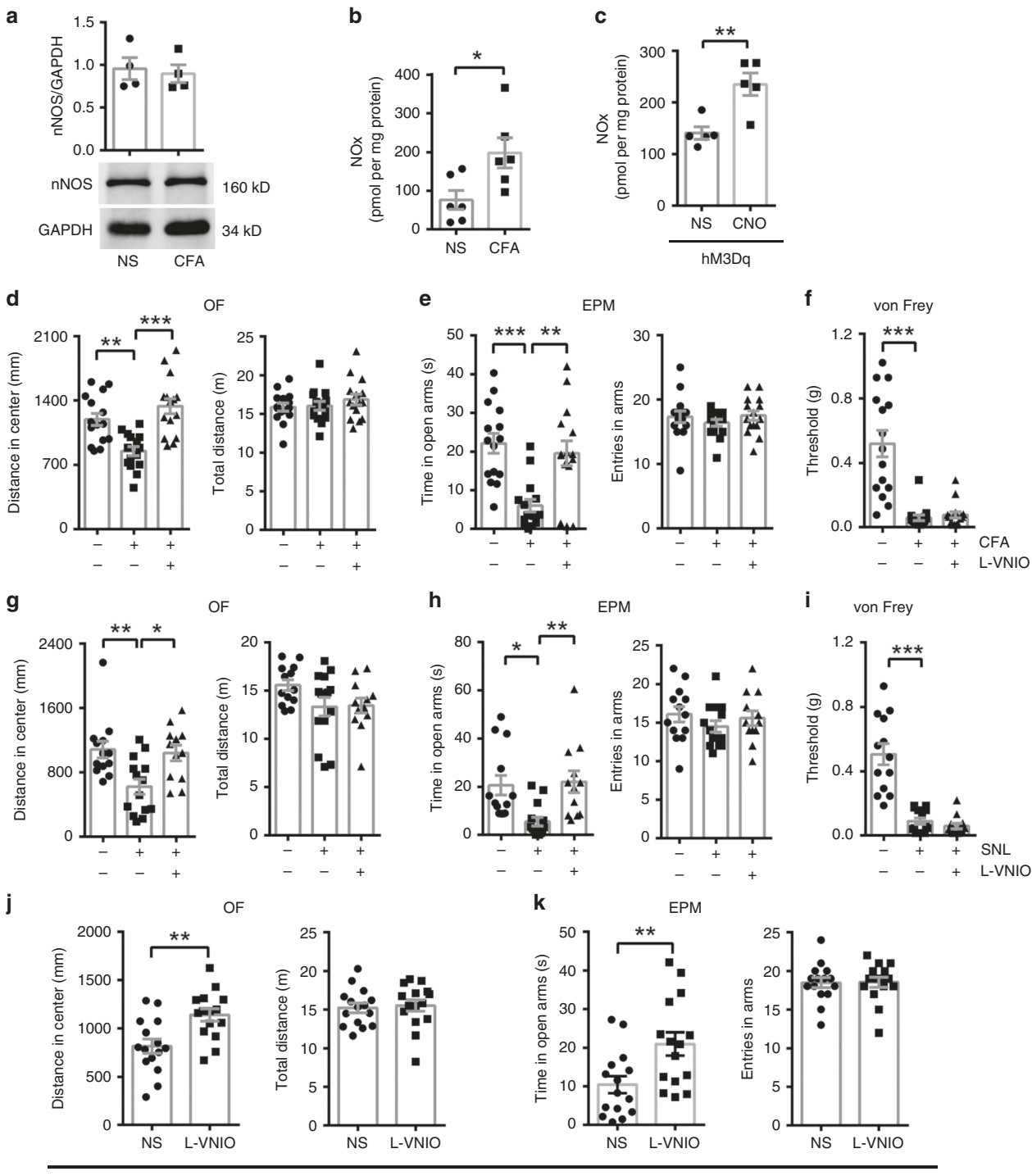

**Fig. 8 nNOS-expressing neurons in the vmPFC mediate chronic pain-induced anxiety through diffusing NO. a**, **b** Dot plot (**a**, upper) and western bolts (**a**, lower) showing nNOS protein levels ($n = 4$), and dot plot (**b**) showing nNOS enzyme activity in the vmPFC at day 3 after CFA or NS injection ($n = 6$). **c** Dot plot showing nNOS enzyme activity of the vmPFC 30 min after CNO (0.5 mM, 1 μl) or NS delivery at day 21 after AAV-hSyn-DIO-hM3Dq-eGFP microinjection (0.4 μl) in the vmPFC of nNOS-Cre mice ($n = 5$). **d–f** Distance in center (**d**, left) and total distance (**d**, right) in an OF test, time in open arms (**e**, left) and total entries in arms (**e**, right) of an EPM test, and withdrawal threshold of hind paw in an von Frey test (**f**) 30 min after L-VNIO (1.5 mM, 1 μl) or NS (1 μl) microinjection into vmPFC at day 3 after CFA or NS injection. $n = 15$. **g–i** Distance in center (**g**, left) and total distance (**g**, right) with an OF test, time in open arms (**h**, left) and total entries in arms (**h**, right) of an EPM test, and withdrawal threshold of the hind paw in an von Frey test (**i**) 30 min after L-VNIO (1.5 mM, 1 μl) or NS (1 μl) microinjection in vmPFC at day 7 after SNL or sham operation. $n = 13$, 14, and 12, respectively. **j**, **k** Distance in center (**j**, left) and total distance (**j**, right) with an OF test, and time in open arms (**k**, left) and total entries in arms (**k**, right) of an EPM test 30 min after L-VNIO (1.5 mM, 1 μl) or NS microinjection in vmPFC. AAV-hSyn-DIO-hM3Dq-eGFP (0.4 μl) was microinjected into the vmPFC, and 3 weeks later, CNO (0.5 mM, 1 μl) was microinjected into the vmPFC 30 min before L-VNIO or NS microinjection. $n = 15$. Data are the mean ± SEM; *$p < 0.05$, **$p < 0.01$, and ***$p < 0.001$ (unpaired two-tailed Student's *t*-test for (**a–c**) and (**j**), (**k**); one-way ANOVA followed by Tukey's *post hoc* test for (**d–i**)). Source data are provided as a Source data file. Exact *p* values and additional statistical information can be found in Source data.

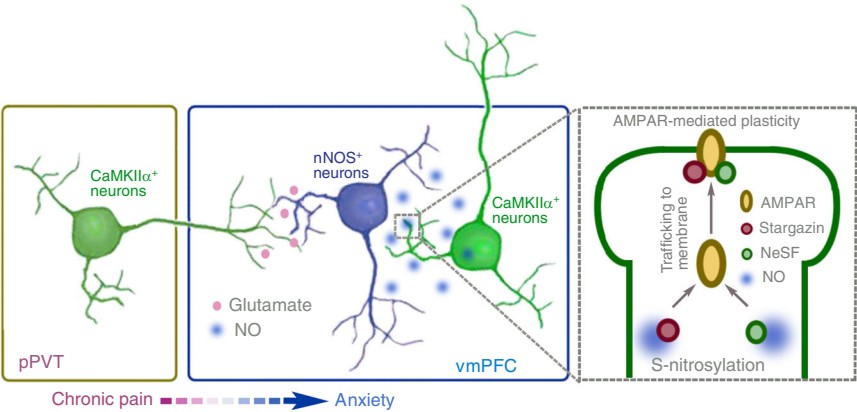

**Fig. 9 A model diagram illustrating the role of vmPFC nNOS-expressing neurons in chronic pain-induced anxiety.** During chronic pain, vmPFC nNOS-expressing neurons are stimulated by glutamatergic inputs from pPVT CaMKIIα⁺ pyramidal neurons. Activated nNOS-expressing neurons produce diffusing NO and cause the S-nitrosylation of AMPAR-interacting proteins in CaMKIIα⁺ pyramidal neurons, promoting AMPAR trafficking to the plasma membrane. Consequently, enhanced AMPAR function in vmPFC CaMKIIα⁺ pyramidal neurons leads to anxiety-related behaviors.

the vmPFC is aroused by chronic pain, which then mediates subsequent anxiety-related behaviors. The mPFC is a critical neuroanatomical hub for controlling motivated behaviors and fear memory, as well as anxiety behaviors; however, the PL and IL are reported to have different and even opposing roles[45,46]. Whether divergent mechanisms under chronic pain-induced anxiety exist in PL and IL remains an open question. Meanwhile, we have not discovered the exact downstream regions innervated by vmPFC neurons in relation to chronic pain-induced anxiety. The previous findings from studies on mPFC circuits are complex. Stimulation of mPFC-nucleus accumbens (NAc) neurons promotes conditioned reward-seeking behaviors, while activity in mPFC-PVT neurons suppresses these behaviors and persistently impairs behavioral cue discrimination[47,48]. There is a time-dependent shift from the PL-basolateral amygdala (BLA) circuit to the PL-PVT circuit in controlling the retrieval of fear memory[49]. IL-amygdala activation but not PL-amygdala activation is anxiolytic, and the basomedial amygdala contributes to this anxiolytic effect, but the BLA does not[46]. It is difficult to speculate which circuit is most likely to mediate chronic pain-induced anxiety. One prominent theoretical perspective suggests that the top-down inhibition of the amygdala by the vmPFC is a crucial neural mechanism that may be defective in certain mood and anxiety disorders[18,46]. Recently, a neural ensemble in the BLA that encodes the unpleasantness of pain has been uncovered[50], providing new evidence to support the critical role of the mPFC-amygdala circuit. However, the considerable body of research challenges the predominant model[14], as described in the "Introduction". Our results, if enhanced AMPAR function necessarily leads to strengthened vmPFC outputs, are contradictory to defective top-down inhibition; they are in agreement with classic pharmacological studies[20–22]. Therefore, the pathology characterizing anxiety disorders cannot be simply described as deactivation of the vmPFC and defective top-down inhibition, particularly in the chronic pain state. There is an interaction between chronic pain and anxiety. The pathological changes in the mPFC of animals and patients with chronic pain may be associated with their anxiety[24,25].

The PVT is thought to be implicated in the regulation of emotional responses, and it broadly projects to the NAc, amygdala, and prefrontal cortex[51,52]. Evidence from optogenetics and chemogenetics suggests that distinct PVT circuits are recruited to modulate different responses. PVT-NAc neurons develop inhibitory responses to reward predictive cues[48], and photoinhibition of anterior PVT-NAc projections increased sucrose seeking when the reward was omitted[53]. The pathway in the PVT-central nucleus of the amygdala is essential for the establishment and expression of fear memory[54], and retrieval at late time points activates these projections[49]. PVT neurons are also activated during visceral nociception[37,55]. We also found that intraplantar injection of CFA induced c-Fos expression in the PVT. PVT-mPFC projections are glutamatergic[56], and they form synapses not only with excitatory pyramidal cells but also with inhibitory interneurons in the mPFC[57]. Enhanced excitation of vmPFC GABAergic neurons by PVT glutamatergic inputs mediates visceral pain[37]. Consistently, we demonstrate that glutamatergic pPVT-vmPFC projections are activated during chronic pain and that their inhibition by treatment with chemogenetics partially reverses mechanical and thermal hyperalgesia. Interestingly, by targeting nNOS-expressing neurons in the vmPFC, the pain signal transmitted through the pPVT-vmPFC circuit induces anxiety-like behaviors. Inhibition of vmPFC nNOS-expressing neurons completely abolishes anxiety-like behaviors shown in OF and EPM tests. Thus, our results demonstrate that pPVT-vmPFC projections are crucial for chronic pain-induced anxiety.

Chronic pain and anxiety disorders contribute to each other's development[4–6]. Common mechanisms underlying chronic pain and chronic pain-induced anxiety are proposed. In PL, the deactivation of excitatory neurons promotes both pain sensation and anxiety[26]. In ACC, the coexistence of two forms of LTP provides a synaptic mechanism for the interactions between chronic pain and anxiety[13]. Of note, chronic pain-induced anxiety is mediated through presynaptic LTP, while postsynaptic LTP plays an important role in only chronic pain but not anxiety[7,12], indicating the divergence of mechanisms underlying chronic pain and induced anxiety. Our results also suggest that there is an anxiety signal derived from but also separated from chronic pain in the vmPFC, which occurs through the activation of nNOS-expressing neurons. Since pPVT-vmPFC projections are involved in algesia, and vmPFC nNOS-expressing neurons are not, it is likely that nNOS-negative neurons in the vmPFC are responsible for the pPVT driving hyperalgesia. This possibility is supported by the fact that there is an obvious tendency of hyperalgesia ($t_{24} = 2.040$, $p = 0.053$, Student's $t$-test) following chemogenetic stimulation of nNOS-negative neurons in the vmPFC (Supplementary Fig. 4c). While further study is needed to identify the exact neuron population, nNOS-expressing neurons have been excluded. Regarding the partial analgesia of pPVT-vmPFC projections inhibition, we suppose that the perception and integration of chronic pain need other more important

structures, such as ACC. This is confirmed by our finding that vmPFC microinjection with the AMPAR antagonist CNQX, which showed analgesic effect when infused in ACC[58], did not distinctly attenuate hyperalgesia.

## Methods

**Animals**. Male young adult (6–7 weeks) C57BL/6 mice (Model Animal Research Center of Nanjing University, Nanjing, China) and nNOS-Cre mice (B6.129-*Nos1*[tm1(cre)Mgmj]/J; The Jackson Laboratory; stock number: 017526) were used in this study. Animals were housed 4–5 per cage, and they were maintained at a controlled temperature (20 ± 2 °C) with 12:12 h light/dark cycle (lights on at 07:00) and ad libitum access to food and water. Animal experiments were designed in accordance with the ARRIVE (Animal Research: Reporting of In Vivo Experiments) guidelines[59], with a commitment to refinement, reduction, and replacement, thus minimizing the number of mice used. All animal experiments were conducted in accordance with the Institutional Animal Care and Use Committee of Nanjing Medical University.

**Chronic pain model**. Chronic inflammatory pain was induced in mice by CFA[13,26,34]. The skin of hindpaw planta was disinfected with 75% ethanol, and 10 μl CFA (1 mg/ml in 85% paraffin oil and 15% mannide monoleate, heat-killed *Mycobacterium tuberculosisin*) was injected subcutaneously into the planta of right hindpaw. Chronic neuropathic pain was induced in mice by SNL[34,35]. Mice were kept anesthetized with 1.5% isoflurane, and placed in a prone position. A median skin incision with about 3–5 cm length in L4–S2 level of the mouse back was made. The right paraspinal muscles were separated from the spinous processes at the L5–L6 levels and the L6 transverse process was carefully removed. L5 nerve was isolated and tightly ligated with 5-0 silk suture. The wound was closed with 4-0 silk suture and covered with iodine solution. Both pain types lead to significant anxiety-related behaviors in rodents[13,26,35,50].

**Anxiety-related behavior tests**. Anxiety-related behaviors of mice in open field (OF), elevated plus maze (EPM), novelty-suppressed feeding (NSF), and light–dark box (LDB) tests were recorded and analyzed[27,28], using TopScan LITE system (Clever Sys Inc., Reston, VA, USA). The experimental procedures were performed in a blind manner. Immediately after each test, the animal was transferred to its home cage. In one experiment, the animals only underwent two behavioral tests (OF and EPM, or NSF and LDB), with an interval of 30 min.

The OF constructed of a plastic box (30 × 30 × 50 cm³) was used as a measure of rodent activity. The center of the OF was aversive and potentially risk-laden, whereas exploration of the periphery provides a safer choice. Anxiety animals displayed more exploration of the periphery in OF test. Mice were placed in a corner of the open field at the beginning of assay, and allowed to explore 5 min. Their behaviors were recorded and analyzed with TopScan LITE software (Clever Sys Inc.). Distance moved in the 15 × 15 cm square region in the center of OF represented the anxiety levels of mice, and total distance moved in OF showed the locomotor activities.

The EPM consists of two risk-laden arms (open without sidewalls, 30 × 5 cm²) and two "safe" arms (closed by sidewalls, 30 × 5 cm², with end and side walls 15 cm high), connected with a central platform (5 × 5 cm²) 50 cm above the floor. Mice were placed in the central platform, with their face pointing toward an open arm and allowed to explore freely for 5 min. Animal behaviors were recorded and analyzed with TopScan LITE software (Clever Sys Inc.). Time spent in the open arms represented the anxiety levels of mice, and total number of entries into four arms showed the locomotor activities.

The NSF test was performed during a 5 min period. In brief, the testing apparatus consisted of a plastic box (50 × 50 × 20 cm³), the floor of which was covered with about 2 cm wooden bedding. Twenty-four hours before behavioral testing, all the food was removed from the home cage. At the time of testing, a single pellet of food (regular chow) was placed on a white paper platform positioned in the center of the box. Each mouse was placed in a corner of the box, facing the corner, and the latency to eat (defined as the mouse sitting on its haunches, holding the pellet with its forepaws, and biting the pellet) was timed. The amount of food consumed by the mouse in 15 min was also measured, serving as a control for change in a petite as a possible confounding factor.

The LDB test was performed with a cage consisting of one light (20 × 30 × 40 cm³) and one dark compartment (20 × 30 × 40 cm³). The test was initiated by placing the mice in the lit side. The mice were allowed to transit freely between sides and the time spent in each compartment was recorded for 4 min. The total distance in two compartments was also recorded, showing the locomotor activities.

**Mechanical and thermal hyperalgesia measurement**. Mechanical hyperalgesia was assessed by measuring the 50% paw withdraw threshold in response to probing using von Frey monofilaments (Touch-Test TM Sensory Evaluator, North coast Medical, Inc., Atlanta, GA, USA)[34]. The mice were placed in a plastic cage (45 cm × 5 cm × 11 cm), which was elevated on a mesh screen. Each mouse was allowed to acclimate for 30 min prior to testing. Von Frey filaments, ranging from 0.02 to 2 g, were used to the plantar surface of the hindpaw until the painful

behavior appeared. The filaments were presented, in ascending order of strength, perpendicular to the plantar surface with sufficient force to cause slight bending against the paw and held for 4 s. The 0.4 g stimulus was applied first. A positive response was noted if the paw was sharply withdrawn. If a positive response was observed, the next smaller Von Frey hair was used; otherwise, the next higher force was applied. The experimental procedures were performed in a blind manner.

Paw withdrawal latency (PWL) in response to thermal hyperalgesia was measured with the Hargreaves test (the Plantar Test Apparatus, Ugo Basile, Italy)[60,61]. Mice were placed individually in a plastic box (17 × 7.5 × 7.5 cm³) on a glass platform. Behavioral accommodation was allowed for 15–20 min. The movable radiant heat source of high-intensity was placed underneath the glass and focused onto the plantar surface of the injured hind paw. To record the PWL in response to thermal stimulus, each mouse was measured 5 times with a 5-min interval, and the mean value (the maximal and minimum value excluded) was recorded. The radiant heat intensity was adjusted at the beginning of the experiment to obtain basal PWLs of 12–15 s, and an automatic 30 s cutoff was used to prevent tissue damage. The experimental procedures were performed in a blind manner.

**Recombinant AAVs and infection**. AAV-CMV-flex-shVGAT-eGFP was generated by GeneChem Co., Ltd. (Shanghai, China) for Cre-dependent knockdown of vesicular GABA transporters (VGAT). The following sequence that produced significant gene silencing was targeted: VGAT, 5′-ACTCATCTTGTGCAATGTATC-3′[43]. AAV-CMV-flex-shVGLUT-eGFP was generated by GeneChem for Cre-dependent knockdown of vesicular glutamate transporters (VGLUT), including VGLUT1, VGLUT2, and VGLUT3. The following sequences that produced significant gene silencing were targeted: VGLUT1, 5′-GCCATGGCATCTGGAGCAAAT-3′[62]; VGLUT2, 5′-GCAAATCTGCTAGGTGCAATG-3′[62]; VGLUT3, 5′-CGGTGGCTTCATTTCAAACAA-3′[42]. AAV-CMV-flex-shControl-eGFP was purchased from GeneChem. Other recombinant viruses used in chemogenetic and optogenetic experiments were commercialized products and purchased from GeneChem or BrainVTA (Wuhan, China). Details are shown in Supplementary Table 1. According to different experiment design, the desired AAV was microinjected into the pPVT (0.2 μl) and/or the vmPFC (0.4 μl) using a stereotaxic instrument for mice (Stoelting, Wood Dale, IL, USA) at a rate of 0.2 nl/s. The parameters were: pPVT: anterior–posterior, −1.3 mm; medial–lateral, 0.0 mm; dorsal–ventral, −3.1 mm. vmPFC: anterior–posterior, +1.7 mm; medial–lateral, ±0.3 mm; dorsal–ventral, −2.8 mm. The needle was withdrawn over a course of 10 min.

The infection rate of each AAV-DREADD was determined with immunofluorescence, using 2–3 mice in the preliminary experiments: AAV-hSyn-DIO-hM3Dq-eGFP, ~83%; AAV-hSyn-DIO-hM4Di-eGFP, ~84%; AAV-CaMKIIα-hM3Dq-mCherry, ~87%; AAV-CaMKIIα-hM4Di-mCherry, ~83%; and AAV-hSyn-DO-hM3Dq-eGFP, ~81%.

**Cannulation and CNO microinjection**. For chemogenetic manipulation of neurons activity, we expressed CNO-based excitatory DREADD (hM3Dq) or inhibitory DREADD (hM4Di) using adeno-associated virus. Cannulation was performed just after the AAVs microinjection (21 days before CNO delivery). Stainless steel guide cannula (26 gauge, RWD Life Science, Shenzhen, China) was implanted into vmPFC (anterior–posterior, +1.7 mm; medial–lateral, ±0.3 mm; dorsal–ventral, −2.2 mm) and fixed to the skull with adhesive luting cement and acrylic dental cement. Following surgery, a stainless steel obturator was inserted into the guide cannula to avoid obstruction until microinjection was made. Mice were briefly head-restrained, while the stainless steel obturator was removed and an injection tube (30 gauge, RWD Life Science) was inserted into the guide cannula. The injection tube was designed to protrude 0.6 mm from the tip of the catheter thus penetrating into the core of vmPFC. At day 21 (without CFA injection) or 24 (CFA injection at day 21) after microinjection of AAVs, CNO was microinjected into vmPFC (0.5 mM, 1 μl) through the implanted cannula to activate DREADD, and the same volume of NS was used as control. CNO or NS was slowly infused at a flow-rate of 0.2 μl per min to a total volume of 1 μl. Following injection, the injection cannula was left in place for 5 min to allow drugs to reduce back-flow. The stainless steel obturator was subsequently reinserted into the guide cannula. Thirty minutes after CNO or NS microinjection, the animal behaviors were tested.

**Immunofluorescence and cell counting**. Immunofluorescent labeling was performed as we previously did[44,63]. Mice were transcardially perfused with 4% paraformaldehyde (w/v) in phosphate buffer (0.1 M, pH 7.4), and brains were removed and post-fixed overnight. Serial vibratome sections (40 μm) were processed. After blocking in phosphate-buffered saline (PBS) containing 3% normal goat serum, 0.3% (w/v) Triton X-100, and 0.1% bovine serum albumin at room temperature for 1 h, slices were incubated in primary antibody diluted in blocking solution overnight at 4 °C and washed. Slices were then incubated in secondary antibody for 2 h at room temperature and counterstained with Hoechst 33258 to label the nuclei. Finally, slices were mounted onto slides and images were acquired with a fluorescence microscope (Axio Imager, Zeiss, Oberkochen, Germany) or a confocal laser-scanning microscope (LSM700, Zeiss). Details for the primary and secondary antibodies used were available in Supplementary Table 1.

Cell counting was conducted on every third section in a series of 40-µm coronal sections throughout vmPFC or pPVT. The fluorescent images were acquired at 10× for c-Fos$^+$ cells counting or at 40× for c-Fos$^+$ and nNOS$^+$ cells counting. The images were analyzed with UTHSCSA ImageTool 3.0 software. To quantify the c-Fos$^+$ cells, the count from each image was divided by the area. The values from all sampled sections of one animal were averaged and the mean was regarded as the final value of one animal. To obtain the ratio of c-Fos$^+$ and nNOS$^+$ cells to nNOS$^+$ cells (c-Fos$^+$ and nNOS$^+$/nNOS$^+$), the counts of c-Fos$^+$ and nNOS$^+$ cells or nNOS$^+$ cells from all sampled sections of one animal were summed and the ratio were calculated.

**Fluorescent in situ hybridization (FISH)**. FISH was performed using RNAscope assay kit (Advanced Cell Diagnostics, Newark, CA, USA). Mice were transcardially perfused with 4% paraformaldehyde (w/v) in phosphate buffer (0.1 M, pH 7.4), and brains were removed and post-fixed overnight. Then the brains were dehydrated by 10, 20, and 30% sucrose gradient solutions in sequence, until the brain completely sink to the bottom. Brain sections (12 µm) were cut in a cryostat (Leica, Germany). All subsequent steps were performed according to the manufacturer's instructions of RNAscope assay kit. Probes used: RNAscope® Probe-Mm-Nos1, RNAscope® Probe-EGFP-C2. Finally, these two probes were visualized by Cy3 and Cy5, respectively. RNAscope® 3-plex negative control probe was used to confirm the specificity of the labeling.

**Western blot**. Western bolt analysis was performed as before[27,44]. Tissues were lysed in 100 mM 4-(2-hydroxyethyl)-1-piperazineethanesulfonic acid (HEPES) containing 200 mM NaCl, 10% glycerol, 2 mM $Na_4P_2O_7$, 2 mM dithiothreitol, 1 mM ethylenediaminetetraacetic acid (EDTA), 1 mM benzamidine, 0.1 mM $Na_3VO_4$, 1 µM pepstatine, 10 µg/ml aprotinin, 10 µg/ml leupeptin, and 10 µM phenylmethylsulfonyl fluoride at pH 7.4. After lysis for 30 min in ice, samples were centrifuged at 12,000$g$, 4 °C for 15 min. The samples containing equivalent amounts of protein were applied to acrylamide denaturing gels (sodium dodecyl sulfate polyacrylamide gel electrophoresis, SDS-PAGE). The separated proteins were transferred onto Immobilon®-P Transfer Membranes (Millipore, Burlington, MA, USA). Blotting membranes were incubated with blocking solution [5% nonfat dried milk powder dissolved in Tris-buffered saline with Tween 20 (TBST) buffer (pH 7.5, 10 mM Tris–HCl, 150 mM NaCl, and 0.1% Tween 20)] for 1 h at room temperature, washed three times, and then were incubated with primary antibodies in TBST overnight at 4 °C. Internal control was carried out using GAPDH antibody. After several washes with TBST buffer, the membranes were incubated for 1 h with appropriate horseradish peroxidase-linked secondary antibody. The membranes were then processed with enhanced chemiluminescence western blotting detection reagents (Bio-Rad, Hercules, CA, USA). The films were scanned with ChemiDOC$^{TM}$ MP Imaging System (Bio-Rad), and densitometry was performed using Image Lab$^{TM}$ software (Bio-Rad). Details for the primary and secondary antibodies used are described in Supplementary Table 1. Uncropped full-length membrane scans for the most important blots are provided in the Source data file.

**nNOS activity assay**. Bilateral vmPFCs were collected and homogenized in ice-cold PBS, pH 7.4, and analyzed using the Nitric Oxide Assay Kit (Beyotime Biotech, Shanghai, China)[64]. Because NO has an extremely short half-life, we quantified NOS activity by measuring the concentrations of the two stable NO products nitrate and nitrite. According to the protocol of the manufacturer, the assay included a process to convert nitrate to nitrite and then to use a Greiss reaction to measure the nitrite concentrations. Absorbance of the samples was measured at 540 nm. To measure nNOS activity, selective eNOS inhibitor diphenyleneiodonium chloride (3 µM, 1 µl, Millipore) and selective iNOS inhibitor 1400 W (100 µM, 1 µl, Sigma-Aldrich) was microinjected into vmPFC through implanted cannula 1 h before the mice were sacrificed for tissue collection.

**Biotin-switch assay**. This assay was performed in the dark[28,63]. Briefly, cells were lysed in HEN buffer (250 mM HEPES, 1 mM EDTA, and 100 mM neocuproine) and adjusted to contain 0.4% CHAPS. Samples were homogenized and free cysteines were blocked for 1 h at 50 °C in three volumes of blocking buffer (HEN buffer plus 2.5% SDS, HENS) containing methyl methanethiosulfonate (200 mM, Sigma-Aldrich). Proteins were precipitated with acetone at −20 °C and resuspended in 300 µl HENS solution. After adding fresh ascorbic acid (20 mM, Sigma-Aldrich) and HPDP-biotin (1 mM, Thermo), proteins were incubated at room temperature for 1 h. Biotinylated proteins were resuspended in 250 µl HENS buffer plus 500 µl neutralization buffer (20 mM HEPES, 100 mM NaCl, 1 mM EDTA, 0.5% Triton X-100) and precipitated with 50 µl prewashed avidin-affinity resin beads (Sigma-Aldrich) at room temperature for 1 h. The beads were washed five times at 4 °C using neutralization buffer containing 600 nM NaCl. Biotinylated proteins were eluted using 30 µl elution buffer (20 mM HEPES, 100 mM NaCl, 1 mM EDTA, 100 mM β-mercaptoethanol) and heated at 100 °C for 5 min in reducing SDS-PAGE loading buffer. Finally, biotinylated proteins were detected by immunoblotting using appropriate antibodies (displayed in Supplementary Table 1).

**Electrophysiology**. Slice preparation for electrophysiology: Mice were sacrificed at day 3 after CFA injection or day 21 after AAVs microinjection and brain slices were prepared[65]. Under deep ethyl ether anesthesia, mice were decapitated. Brains were quickly removed and placed into ice-cold buffer containing 110 mM choline chloride, 20 mM glucose, 2.5 mM KCl, 0.5 mM $CaCl_2$, 7 mM $MgCl_2$, 1.3 mM $NaH_2PO_4$, 25 mM $NaHCO_3$, 1.3 mM Na-ascorbate, 0.6 mM Na-pyruvate. Coronal brain slices (350 µm) containing either vmPFC or pPVT were prepared using a vibratome, and the slices were incubated in an interface-style chamber containing normal artificial cerebrospinal fluid (aCSF) composed of 10 mM glucose, 125 mM NaCl, 2.5 mM KCl, 2 mM $CaCl_2$, 1.3 mM $MgCl_2$, 1.3 mM $NaH_2PO_4$, 25 mM $NaHCO_3$, 1.3 mM Na-ascorbate, 0.6 mM Na-pyruvate at 25 °C for at least 1 h. All solution was gassed with 95% $O_2$ and 5% $CO_2$. All the following electrophysiological experiments were performed in a blind fashion.

**Spontaneous EPSCs recording**. sEPSCs were recorded by whole-cell patch-clamp[26]. CaMKIIα-mCherry pyramidal neurons in vmPFC were visualized using a fluorescent and an infrared-DIC optical system combined with a CCD camera and monitor, and those in layer 2/3 were recorded because their activity is highly related with anxiety behaviors[26,66,67]. Patch electrodes (4–6 MΩ) were filled with pipette solution contained (in mM): 132.5 Cs-gluconate, 17.5 CsCl, 2 $MgCl_2$, 0.5 ethylene glycol bis(2-aminoethyl)tetraacetic acid (EGTA), 10 HEPES, 4 ATP, 5 QX-314 (pH 7.3). For voltage-clamp recordings, sEPSCs were recorded at −60 mV. All recordings were performed in the presence of 20 µM BMI and 50 µM AP-5 to isolate AMPAR-mediated currents. In the experiment of imitating pathology via DREADD, CNO (5 µM) was used to excite neurons expressed hM3Dq and C-PTIO (10 µM) was used to eliminate the NO produced in vmPFC. After the slices were incubated for 30 min, BMI and AP-5 were added to the aCSF for recording. Amplitude and frequency of sEPSCs were analyzed with Mini software (Synaptosoft Inc., Fort Lee, NJ, USA).

**Miniature EPSCs recording**. Visualized patch-clamp recordings from layer 2/3 pyramidal neurons in vmPFC were performed at 40× using infrared oblique-illumination (Olympus X51W; Hamamatsu CCD camera C11440)[65]. Microelectrodes (6–8 MΩ) were filled with internal pipette solution, containing 132.5 mM Cs-gluconate, 17.5 mM CsCl, 2 mM $MgCl_2$, 0.5 mM EGTA, 10 mM HEPES, 4 mM ATP, 5 mM QX-314. To study mEPSC activity in isolation, tetrodotoxin (0.5 µM) and BMI (20 µM) were added to block action potentials and GABA$_A$ receptor-mediated currents, respectively. The recording was more than 5 min. Data were analyzed using Mini software. Up to 100 events from each neuron were selected at a fixed sampling interval to generate cumulative probability. All recordings were low-pass filtered at 2 kHz and acquired at 10 kHz. Cells in which the $R$ or capacitance deviated by 20% from initial values, or $R > 20$ MΩ at any time during the recording were excluded from the analysis.

**Evoked EPSCs recording**. eEPSCs recording in Fig. 4: ChR2-eYFP axon terminals from pPVT were located within vmPFC, and mCherry$^+$ nNOS-expressing neurons surrounded by these axons were then visualized via infrared oblique-illumination (Olympus X51W; Hamamatsu CCD camera C11440) and randomly selected for voltage-clamp recordings. Patch pipettes were made from borosciliate glass using a micropipette puller (Model P-1000; Sutter Instruments). Pipette solution contained (in mM): 132.5 Cs-gluconate, 17.5 CsCl, 2 $MgCl_2$, 0.5 EGTA, 10 HEPES, 4 ATP, 5 QX-314 (pH 7.3). Pipettes with resistances ranging from 4 to 6 MΩ were selected for recording. Evoked excitatory post-synaptic currents (eEPSCs) were recorded as reported[14] by voltage clamping at −65 mV using an Axonpatch-700B amplifier (Axon Instruments). To study glutamate receptors-mediated eEPSC activity in isolation, BMI (20 µM) was added to block GABA$_A$ receptors-mediated currents. For optogenetic stimulation of ChR2 channels, light was delivered to the slice through an optical fiber from a 465 nm wavelength blue laser. Single light pulse (5 ms duration) delivered to activate pPVT fibers in vmPFC while recording light-evoked EPSCs in nNOS-expressing neurons. In this experiment, CNQX (10 µM) and AP-5 (50 µM) were used to block AMPARs- and NMDARs-mediated EPSCs and TTX (0.5 µM) was used to block APs. 4-AP (100 µM) was used to rescue TTX-blocked EPSCs. All recordings were low-pass filtered at 2 kHz and acquired with a Digidata 1440A at 10 kHz (Axon Instruments). Cells in which the $R$ or capacitance deviated by 20% from initial values, or $R > 20$ MΩ at any time during the recording were excluded from the analysis. Data were collected with pClamp 10.3 software and analysis using Clamfit 10.3 (Molecular Devices).

eEPSCs recording in Fig. 7 and Supplementary Fig. 11: ChR2-eYFP$^+$ nNOS-expressing neurons in vmPFC were stimulated by blue light and mCherry$^+$ (Supplementary Fig. 11) or eYFP$^-$ (Fig. 7) pyramidal neurons in layer 2/3 of vmPFC were randomly selected for voltage-clamp recordings. Method details were the same as eEPSCs recording in Fig. 4, except that all recordings were performed in the presence of TTX (0.5 µM) and 4-AP (100 µM) to obtain monosynaptic transmission.

AMPA/NMDA ratio in Figs. 5 and 6: For AMPA/NMDA ratio experiments, aCSF was supplemented with 20 µM BMI to block inhibition. AMPA receptors-mediated EPSCs were calculated by averaging 15 EPSCs at −70 mV and measuring the peak compared with the baseline. NMDA receptors-mediated EPSCs were

calculated by averaging 15 EPSCs at +40 mV and measuring the amplitude 50 ms after the stimulation compared with the baseline.

**Evoked IPSCs recording**. mCherry[+] (Supplementary Fig. 11) or eYFP[−] (Fig. 7) pyramidal neurons in layer 2/3 of vmPFC were visualized via infrared oblique-illumination (Olympus X51W; Hamamatsu CCD camera C11440) and randomly selected for voltage-clamp recordings. Patch pipettes (4–6 MΩ) were made from borosciliate glass using a micropipette puller (Model P-1000; Sutter Instruments). These pipettes were filled with pipette solution contained (in mM): 132.5 Cs-gluconate, 17.5 CsCl, 2 MgCl$_2$, 0.5 EGTA, 10 HEPES, 4 ATP, 5 QX-314 (pH 7.3). Evoked inhibitory post-synaptic currents (eIPSCs) were recorded by voltage clamping at +10 mV[68] using an Axonpatch-700B amplifier (Axon Instruments). For optogenetic stimulation of ChR2-eYFP[+] nNOS-expressing neurons in vmPFC, Single light pulse (5 ms duration) was delivered to the slice through an optical fiber from a 465 nm wavelength blue laser. All recordings were performed in the presence of TTX (0.5 μM) and 4-AP (100 μM) to obtain monosynaptic transmission.

**Action potentials recording**. To confirm the functionality of expressed hM3Dq and hM4Di, APs of hM3Dq or hM4Di expressing neurons in vmPFC were recorded by whole-cell current-clamp[69]. pipettes (4–6 MΩ) were filled with electrode internal solution composed of (in mM) 70 K-gluconate, 70 KCl, 2 NaCl, 2 MgCl$_2$, 10 HEPES, 1 EGTA, 2 MgATP, and 0.3 Na$_2$GTP (pH 7.3). In these experiments, an appropriate current was injected into the patched neurons to induce spiking. And then, after 3 min continuous recording, CNO (5 μM) was added to aCSF for the following recording.

**Quantification and statistical analysis**. Appropriate parametric statistics were utilized to test our hypothesis. Comparisons among multiple groups were made with one-way ANOVA followed by Tukey's *post hoc* test (GraphPad Prism 6 software). Tukey's *post hoc* test was performed only when ANOVA yielded a significant main effect. Comparisons between two groups were made with unpaired two-tailed Student's *t*-test (GraphPad Prism 6 software). Cumulative probability plots were produced for the analysis of sEPSCs and mEPSCs (Mini Analysis software 6.0), and statistical analysis was performed on mean values. Data are presented as the mean ± SEM, and $p < 0.05$ was considered statistically significant. Statistical details of the experiments are shown in "Results", figures, and figure legends.

**Reporting summary**. Further information on research design is available in the Nature Research Reporting Summary linked to this article.

## Data availability
All the data supporting the findings of this study are available within the paper and its Supplementary Information file. The source data underlying all figures are provided as a Source data file.

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

## Acknowledgements

This work was supported by grants from the National Natural Science Foundation of China (81571188 and 31530091), the National Key R&D Program of China (2016YFC1306703), the Science and Technology Program of Guangdong (2018B030334001), the Southeast University–Nanjing Medical University Joint Fund (2242017K3DN10), and the Training Program for Distinguished Young Scholar of Nanjing Medical University (2017NJMURC003). The authors thank T.F. Ma for helpful advice about electrophysiological experiments and K. Xu, Y.X. Song, Y. Tao, and Y. Zhou for technical assistance with behavior tests.

## Author contributions

H.-Y.L., Z.-J.C., H.X., and Y.-Y.H. carried out and analyzed the behavioral tests. H.-Y.L., Z.-J.C., H.X., and Y.-H.L. performed and analyzed the electrophysiological experiments, fluorescent imaging experiments, and biochemical assays. L.C. and H.-Y.W. prepared chronic pain models and conducted cannulations and microinjections. W.L. directed the electrophysiological experiments. C.-X.L., D.-Y.Z., and P.W. conceived the study. C.-X.L. designed and directed the experiments and wrote the manuscript.

## Competing interests

The authors declare no competing interests.
