## [Peer Review File · Nature Communications]

Reviewers' comments:

Reviewer #1 (Remarks to the Author):

In their manuscript entitled "nNOS-expressing neurons in vmPFC transduce PVT-derived chronic pain signals to anxiety behaviors" Prof. Luo and colleagues seek to investigate the role of a subpopulation of vmPFC neurons, those that express neuronal nitric oxide synthase (nNOS), in the affective effects of chronic pain. Using an intersectional approach that includes animal models of both inflammatory and neuropathic pain, chemogenetics, optogenetics and slice recordings the authors suggest that a thalamic input from the PVT specifically drives vmPFC nNOS neuronal activity thereby promoting anxiety-related behaviors. In addition, the authors propose that this process requires nNOS-mediated glutamatergic plasticity in the vmPFC. However, while at first glance the data seems to hit all its intended targets, in depth assessment reveals that a series of critical controls, quantitative analyses and experimental manipulations are missing. Instead, many conclusions are drawn without empirical evidence or an adequate experimental design. As such, in its current version, the manuscript does not provide sufficient support for the authors' claims. However, if after a thorough and extensive experimental revision of their manuscript the authors provide convincing support to their claims, their study should be considered for publication.

Below is a summary with specific comments:

- 1) The authors essentially base their interest on studying the function of nNOS+ neurons of the vmPFC in the context of pain from a c-Fos experiment presented in Fig. 1f as well as supplementary Fig. 2. However, there is no information in the manuscript about how this data was quantified. In addition, from the sample image presented is not even clear if there are any pain-induced increases in c-Fos expression in the vmPFC. This is important considering the authors' later claims that glutamatergic plasticity and increase vmPFC neuronal activity underlie the affective effects of chronic pain. The authors should quantify total cFos expression in vmPFC and assess the percentage of nNOS neurons activated. Also, in addition to the merged image, please show cFos and nNOS channels separately.
- 2) The authors should validate their nNOS-Cre mouse as well as the antibody they used. The idea of these "lightly labeled type II nNOS" neurons seems problematic without further support validating that they are indeed nNOS+. For instance, it could be that many real nNOS+ cells are missed from the cFos quantification as a result of poor labeling. Using a technique like multiplexed in situ hybridization would allow the authors to determine the specificity of their manipulations.
- 3) Are DREADDs manipulations unilateral or bilateral? This is not clear in the manuscript. More importantly, the authors use excitatory DREADD (hM3Dq) to argue that activation of these neurons increases anxiety. However, it is unclear whether this phenotype is exclusive for nNOS+ cells. To test this, the authors should investigate whether Cre- neurons of nNOS-Cre mice are equally important. This is a critical issue, because there is an increasing sense that the authors have chosen to focus on these nNOS neurons without sufficient scientific evidence in favor of a selective role of this neuronal population.
- 4) A major issue in this study is the lack of sufficient control for some of the most important chemogenetic experiments. For instance, in Fig. 1q-u the authors want to argue that vmPFC nNOS+ neuron activity is required for the affective component of pain. Specifically, they showed that chemogenetic inhibition of these neurons "rescues" the affective phenotype induced by CFA. However, a CNO-only group is missing from the hM4Di group. This is a major problem because it is unclear whether that manipulation alone is anxiolytic. If it is, then the presumed rescue of the CFA-induced

behavior could simply be due to masking by the anxiolytic effect of nNOS+ neuronal silencing.

5) A couple of recent studies suggest that the anterior PVT is involved in hyperalgesia. However, in the present manuscript the authors seem to focus on the posterior PVT. What is the rationale for this? Is it the case that both anterior and posterior PVT are similarly implicated in pain?

6) Unlike manipulations of vmPFC nNOS+ neurons, chemogenetic excitation and inhibition of PVT-vmPFC projections drive and decrease hyperalgesia, respectively. Is PVT driving hyperalgesia through a separate vmPFC node? Similar to Figure 1 a CNO-only control is missing from the hM4Di experiment. Thus, it is unclear if silencing PVT input to the vmPFC opposes or simply masks CFA-induced hyperalgesia.

7) In Figure 3 the authors look at cFos expression to determine if more nNOS+ neurons are active following activation of PVT input with hM3Dq and CNO than in the control (NS) group. The authors should also show what is the overall increase in cFos expression in the vmPFC following PVT input activation and what proportion of these neurons are actually nNOS+. Also, it appears that the lack of effect in Figure 3m is inconsistent with data from Figure 2. Can the authors comment on this?

8) It isn't totally clear why the authors opted to measure sEPSCs as opposed to mEPSCs to assess glutamatergic transmission in vmPFC. Though the authors show that CFA treatment is associated with increased amplitude of sEPSCs in the vmPFC, it is possible that the effect is mediated by increased network activity. The authors should measure mEPSCs to investigate whether AMPA-mediated CFA-induced plasticity can be observed electrophysiologically. Other measures of AMPA-plasticity such as AMPA/NMDA ratio would be extremely valuable. Also, having assessed this, the authors need to specifically manipulate nNOS+ neurons during CFA to investigate whether nNOS+ neuronal activity is required for the putative glutamatergic plasticity.

9) The authors provide evidence that chemogenetic activation of nNOS+ neurons in vitro, increases sEPSC amplitude and suggest that CFA likely engages similar "cellular" mechanisms. However, if CFA and the chemogenetic activation of nNOS+ neurons act through similar mechanisms, then CFA treatment should occlude the effect of the in vitro chemogenetic manipulation. The authors must investigate this experimentally.

10) Similar to previous experiments where a CNO-only control was missing for hM4Di manipulations, the authors did not include an important control for data shown in Fig. 6d-h. In this figure, the authors show that an NOS specific inhibitor L-VNIO rescues the anxiogenic behavior induced by CFA, but the authors do not include an L-VNIO-only control to determine whether this manipulation alone is sufficient to alter affective behavior. This needs to be addressed experimentally.

Reviewer #2 (Remarks to the Author):

Liang and colleagues use an elegant array of behavioral experiments coupled with cell-type and projection-specific DREADD manipulations to show that PVT inputs to PFCnNOS neurons are activated (Fos) by chronic pain assays and underlie anxiety states that are associated with those assays. Additionally, they show that PFCnNOS neurons likely induce those anxiety states by increasing AMPAR signaling in downstream PFCaMK2+/nNOS- excitatory neurons in layers II/III of the mPFC. The experiments are elegant, with only a few missing controls that are addressable with experiments

and/or acknowledgement within the paper. One major issue is that I found the writing to be somewhat subpar, with substantial improvement needed to paint the story that I described in the few sentences above. However, I am confident that the authors can provide such revisions. I am happy to help if the authors would like additional assistance with any of the writing. This paper should be published in Nature Communications after major revisions. – James M. Otis, Medical University of South Carolina

Major Concerns

1. My biggest concern of the paper is related to the story of neural circuitry as currently described in the paper. To be clear, the experiments paint an amazing picture of how the PVT could influence nNOS-expressing neurons during chronic pain, causing downstream NO signaling in CaMKII-expressing excitatory neurons in layer II/III mPFC which may induce a state of anxiety that is commonly associated with chronic pain. However, it took me the entire paper to figure out that this is the exact circuitry that the authors found. In my opinion, the authors need to clearly lay out the circuitry in the Abstract, Introduction, discuss how it was tested, and discuss how the circuitry is likely working to control chronic pain-induced anxiety. Additionally, in the Results the authors need to be much clearer about what each particular experiment means for the overall circuit involved. A figure at the end showing PVTPFCnNOSPFCCaMK2a+/nNOS- is something they could consider. The first few sentences of the discussion should also more clearly describe the full circuit that I just mentioned.

Minor Concerns

1. My second, more minor concern is related to the control treatments used for behavioral studies. For example, in Figure 2 CNO is not given to a control mouse expressing just mCherry instead of hm3Dq (Fig 2e-i) or hm3Di (Fig 2l-p). CNO is known to have off-target effects, and thus it is important to include a control group to show that behavioral results are not related to CNO treatment alone. I would prefer to have this control group for all experiments, but I suppose adding it to just one, revealing that CNO alone does not influence the studied behaviors, would be sufficient. The authors could also probably indicate that CNO induced opposing behaviors in excitatory DREADD and inhibitory DREADD conditions, suggesting no non-specific effects, but this needs to be very clear in the results sections and possibly even in the discussion.

2. Being someone from outside of the chronic pain and anxiety fields, I'm confused as to why CFA and von Frey hair stimulation were used. At the very least, references should be used when they are first brought up to indicate other papers that have validated these methods supporting their use as a model system for anxiety and chronic pain.

3. I find that the authors often overstep conclusions made regarding neural circuit function for a behavior. For example, "...mice exhibited mechanical hyperalgesia..." does not seem entirely supported by the experiments. Instead, the mice expressed behaviors that are consistent with hyperalgesia, or, the experiments suggest that mice were expressing mechanical hyperalgesia. I don't believe the authors have the data to know how the animals felt, unless I'm missing something about how algesia is monitored. The same is true for conclusions made about neural circuit function for the behaviors studied. Just because we as scientists can bidirectionally modify behavior with CNO, and measure Fos activity, does not mean those circuits causally control those behaviors in natural conditions. Be careful to choose phrasing in a way that accounts for these possibilities, by either explaining more thoroughly or adding words like "suggests" or "indicates", especially in the results section before the final conclusions for the paper are made.

4. Degrees of freedom should be listed for all analyses. T-tests should be described as independent or dependent before judgment can be made about whether the analyses were performed appropriately or not.

5. The "vmPFC" placement is in both prelimbic and infralimbic. Although infralimbic is historically considered part of vmPFC, most researchers do not consider prelimbic to be a part of vmPFC (see Laubach et al., 2018 eNeuro; PMID: 30406193 for a descriptive analysis of this). Thus, the authors likely need to take out the "v" in "vmPFC", unless they can provide further explanation and citations for why they feel vmPFC is the most appropriate nomenclature. The two citations currently listed (ref #14, 15 in text), are not really appropriate for this claim and should be removed.

6. Related to the comment above, further clarification about placements would be useful. The injection site, which is at -2.2DV from Bregma, should hit prelimbic but not infralimbic according to the Paxinos and Watson mouse brain atlas – especially considering that viruses spread mostly upwards, not downwards, after injection. Have the researchers confirmed that both prelimbic and infralimbic cortices have been targeted for all experiments?

7. The usefulness of Figure 2D and 2K are limited, given that CNO is infused at the terminal projections, not cell bodies. If the terminals do not contain the G-protein machinery required for CNO to work, then somatic but not terminal inhibition would work. Regardless, if CNO only has a behavioral effect in DREADD-injected animals only (see Major Concern #1), this would mitigate this concern substantially.

8. CAPON should be both described and given full name on page 4 before abbreviation. i.e., what is a partner protein?

9. CFA, NS, SNL, and all other acronyms should be both described and given full names and descriptions before abbreviations.

10. It would be helpful if abbreviations in Figures were listed at end of Figure Legend.

11. Some images in the figures are very oversaturated (e.g., Fig 1F and 1J), and thus contrast should be adjusted to allow visualization of the real fluorescence profile of the visualized tissues.

12. Figure 4: C-PTIO is written as C-PITO.

13. Figure 4: A C-PTIO control group that does not receive CNO should have been included, or the conclusion cannot be made that the elevation in sEPSC amplitude is NO dependent.

14. Figure 4 Legend: It would be helpful if the authors could indicate that the trafficking of AMPARs is in layer II/III excitatory pyramidal neurons as in Figure 5 legend.

15. Sentences and Headings such as "nNOS-expressing neurons in vmPFC mediates chronic pain-induced anxiety through the nNOS-NO pathway" could be somewhat misleading. Make it clear that this is happening through NO signaling in downstream CaMK2a-expressing neurons, not specifically in the nNOS-expressing neurons.

16. Much of the discussion seems to be reiteration, even at verbatim (see paragraph 2), of what was said in the introduction. Discussion should extend, rather than reiterate, what has been said in the introduction.

17. This is more of personal preference since I'm a circuit neuroscientist, but I think the authors could spend more time discussing the circuits involved. For example, the different PVT projections that have been implicated in anxiety, fear, and motivation (PVTCEA, PVTPFC, PVTNAC; Greg Quirk Lab, Bo Li Lab, Stuber Lab, Xiaoke Chen Lab). I would also really like to hear more about how PFC outputs from these layer II/III CamK2 neurons might influence those anxiety states. For example, PFCNAC, PFCPVT, PFCBLA circuits have been implicated in anxiety, fear, and/or motivated behaviors (see the same labs, also Peters, Kalivas, Quirk 2009 Learning and Memory for review).

18. There are some easily-fixed grammar and spelling errors that should be addressed before publication.

Reviewer #3 (Remarks to the Author):

The manuscript by Liang and colleagues applies mouse models of pain and slice electrophysiology along with opto- and chemogenetics to investigate the impact of a NOS circuit in the mPFC on anxiety-like and sensory symptoms of chronic pain. The investigators report that chronic pain states affect glutamatergic PVT-vmPFC inputs, and then S-nitrosylation of AMPARs interacting proteins by increased nNOS enzyme activity and NO production which AMPARs trafficking to plasma membrane.

This concept is not particularly novel, as the role of the prefrontal cortex in responses to stress or pain-related anxiety has been reported by many investigators. Most of these studies, are not cited in this manuscript. The novelty here involves the focus on specific nNOS cellular populations, whereas the proposed molecular mechanism further supports the well characterized synaptic adaptations in the glutamatergic system and the cell surface expression of AMPA receptors.

The characterization of NOS expressing neurons and circuit analysis involve an early time point after the induction of peripheral inflammation. First of all, I am concerned about the selection of the CFA model for the study of pain-induced anxiety. This peripheral inflammation model leads to spontaneous recovery from sensory hypersensitivity within a few weeks, whereas nerve injury models promote persistent pain states and long-term anxiety. It is surprising that the investigators only observe short term anxiety, as in murine models of chronic pain anxiety-depression like behaviors persist after recovery from mechanical allodynia. Additional models should be used (findings using the SNL model are only reported in supplementary information) to better characterize this circuit. The behavioral analysis is limited to two tests for anxiety-related behaviors, open field and EPM, and to the use Von Frey assays for the assessment of mechanical allodynia. In the current model, mechanical allodynia is observed for several weeks, whereas anxiety is only observed during the first two weeks after CFA treatment. If the goal of the study is to highlight the role of the particular PVT-PFC circuit on pain-induced anxiety, additional behavioral assays should have been included. A wider range of behavioral paradigms reflecting sensory and affective symptoms, as well as motivational states are required in order to better assess the impact of the PFC NOS neurons and the PVT-PFC projections on chronic pain symptomatology.

Other comments

-Western blot analysis studies: Figures show representative blots, a larger amount or samples or the full membrane along with loading controls should be provided.

-Several statements are confusing, mostly due to language issues.

Reviewer #1

1) The authors essentially base their interest on studying the function of nNOS⁺ neurons of the vmPFC in the context of pain from a c-Fos experiment presented in Fig. 1f as well as supplementary Fig. 2. However, there is no information in the manuscript about how this data was quantified. In addition, from the sample image presented is not even clear if there are any pain-induced increases in c-Fos expression in the vmPFC. This is important considering the authors' later claims that glutamatergic plasticity and increase vmPFC neuronal activity underlie the affective effects of chronic pain. The authors should quantify total cFos expression in vmPFC and assess the percentage of nNOS neurons activated. Also, in addition to the merged image, please show cFos and nNOS channels separately.

We are sorry to omit the method for quantitative analysis of nNOS⁺&c-Fos⁺/nNOS⁺ ratio. The detail has been described in the revised manuscript. Please see Supplementary Information (page 7, paragraph 2).

As the reviewer pointed out, c-Fos expression in the vmPFC is important. So we also quantified the c-Fos⁺ cells as well as the percentage of nNOS neurons activated (please see Figure 1e; text on page 6, line 19-21). The same as the nNOS⁺&c-Fos⁺/nNOS⁺ ratio increased, the c-Fos⁺ cells in vmPFC of CFA-treated mice was more than control. This result was consistent with our later claims that glutamatergic plasticity and increase vmPFC neuronal activity underlie the affective effects of chronic pain. Please note that c-Fos⁺ cells were quantified as positive cells number/mm² in the revised manuscript. So we converted the absolute number of c-Fos⁺ cells to positive cells number/mm² in Figure 3a (Figure 2a in the 1st version).

In addition, we have shown c-Fos and nNOS channels separately in the revision (Figure 1d). From the c-Fos-labeled channel, it is clearer than from the merged image to see pain-induced increase in c-Fos expression in the vmPFC.

2) The authors should validate their nNOS-Cre mouse as well as the antibody they used. The idea of these "lightly labeled type II nNOS" neurons seems problematic without further support validating that they are indeed nNOS⁺. For instance, it could be that many real

nNOS⁺ cells are missed from the cFos quantification as a result of poor labeling. Using a technique like multiplexed in situ hybridization would allow the authors to determine the specificity of their manipulations.

We strongly agree with the reviewer that we should validate the nNOS-Cre mice as well as the antibody we used. Indeed, we had done this work by double-labeling nNOS with Cre-dependent eGFP with immunofluorescence (Figure 1i), which is widely used to validate the efficacy and specificity of Cre-Loxp system (Leshan et al., *Nat Med.* 2012 May;18(5):820-3; McCall et al., *Neuron.* 2015 Aug 5;87(3):605-20; Bocchio et al., *eNeuro.* 2016 Oct 5;3(5). pii: ENEURO.0177-16.2016). Moreover, we purchased this strain of nNOS-Cre mice from The Jackson Laboratory (B6.129-*Nos1^{tm1(cre)Mgmj}/J*; stock number: 017526), and many researches have been reported with this strain (Zimmerman et al., *Nature.* 2019 Apr;568(7750):98-102; Jelitai et al., *Nat Commun.* 2016 Dec 15;7:13722; Augustine et al., *Nature.* 2018 Mar 8;555(7695):204-209). I know the reviewer's main concern is whether the lightly labeled type II nNOS neurons are really nNOS⁺ neurons. Classically, nNOS-expressing neurons are subdivided into two types according to the intensity of NOS/NADPH staining: heavily labeled type I neurons and weakly labeled type II cells (Perrenoud et al., *Front Neural Circuits.* 2012 Jun 29;6:36; Magno et al., *Front Neural Circuits.* 2012 Sep 24;6:65; Hashikawa et al., *Brain Res.* 1994 Apr 4;641(2):341-9; Kubota et al., *Cereb Cortex.* 2011 Aug;21(8):1803-17). Moreover, we validated the nNOS antibody used in this study with nNOS KO mice (Supplementary Figure 3). So we are convinced that the lightly labeled type II nNOS neurons are really nNOS⁺ neurons and Cre-dependent expression are well co-localized with nNOS.

All the same, we performed multiplexed in situ hybridization to further support the validity of nNOS-Cre mice in the revised manuscript, as the reviewer's suggestion. The result was shown in Supplementary Figure 3c-d.

3) Are DREADDs manipulations unilateral or bilateral? This is not clear in the manuscript. More importantly, the authors use excitatory DREADD (hM3Dq) to argue that activation of these neurons increases anxiety. However, it is unclear whether this phenotype is exclusive for nNOS⁺ cells. To test this, the authors should investigate whether Cre- neurons of nNOS-Cre

mice are equally important. This is a critical issue, because there is an increasing sense that the authors have chosen to focus on these nNOS neurons without sufficient scientific evidence in favor of a selective role of this neuronal population.

According to the reviewer's suggestion, we have included an additional experiment in the revision to address this issue. AAV-hSyn-DO-hM3Dq-eGFP were microinjected into the vmPFC of nNOS-Cre mice and CNO was given to stimulate nNOS⁻ (Cre⁻) neurons. Behavioral tests showed that activation of nNOS⁻ neurons in vmPFC did not cause anxiety-like behaviors, indicating the exclusive role of nNOS⁺ neurons in anxiety regulation. Please see Supplementary Figure 4a-e and text on page 8, line 2-6.

Additionally, DREADDs manipulations are bilateral. We had described this in Supplementary Methods. Please see the stereotactic coordinates in "Recombinant AAVs and infection" and "Cannulation and CNO microinjection". To make it clearer to readers, we stated bilateral manipulations in Materials and Methods section of the text in the revised manuscript (page 26, line 9 and 17).

4) A major issue in this study is the lack of sufficient control for some of the most important chemogenetic experiments. For instance, in Fig. 1q-u the authors want to argue that vmPFC nNOS⁺ neuron activity is required for the affective component of pain. Specifically, they showed that chemogenetic inhibition of these neurons "rescues" the affective phenotype induced by CFA. However, a CNO-only group is missing from the hM4Di group. This is a major problem because it is unclear whether that manipulation alone is anxiolytic. If it is, then the presumed rescue of the CFA-induced behavior could simply be due to masking by the anxiolytic effect of nNOS⁺ neuronal silencing.

To address the issue of lacking CNO-treated hM4Di control group in Figure 1q-u of original manuscript (Figure 2a-d in revision), we added an experiment in the revision to observe whether chemogenetic inhibition of vmPFC nNOS⁺ neurons alone is anxiolytic (Supplementary Figure 5, text on page 8, line 12-14). As the data shown, this manipulation only had an anxiolytic tendency without significance. Taken together, we can claim that chemogenetic inhibition of nNOS⁺ neurons in vmPFC rescues the affective phenotype induced by CFA. Moreover, this experiment can also be regarded as the non-strict control for

Figure 2e-h.

5) A couple of recent studies suggest that the anterior PVT is involved in hyperalgesia. However, in the present manuscript the authors seem to focus on the posterior PVT. What is the rationale for this? Is it the case that both anterior and posterior PVT are similarly implicated in pain?

According to *The Mouse Brain in Stereotaxic Coordinates*, Third Edition (Keith B.J. Franklin & George Paxinos, Academic Press of Elsevier), PVT is composed of three subregions: PVA (paraventricular thalamic nucleus, anterior part), PV (paraventricular thalamic nucleus) and PVP (paraventricular thalamic nucleus, posterior part). We focused on the middle part PV. The two recent studies on PVA mentioned by the reviewer, I think, are *Pain*. 2019 May;160(5):1208-1223 and *J Neurosci*. 2010 Aug 4;30(31):10360-8. They are from the same lab, and only investigate chronic pain itself without the related anxiety behaviors. Anatomically, the PVA is connected to regions involved in pain perception, including the anterior cingulate cortex (ACC), central nucleus of amygdala (CeA), parabrachial (PB) area, and periaqueductal gray (PAG) (Chang et al., *Pain*. 2019 May;160(5):1208-1223). So it is an attractive region for pain research. Why we chose to focus on the middle part of PVT is that its projection to mPFC is reported to be involved in affective components of visceral nociception (Jurik et al., *Pain*. 2015 Dec;156(12):2479-91) and we wanted to seek the region which can innervate vmPFC nNOS-expressing neurons when chronic pain induces anxiety. Together, we suppose that the whole PVT is implicated in modulation of chronic pain, but different subregions might have different functions.

6) Unlike manipulations of vmPFC nNOS⁺ neurons, chemogenetic excitation and inhibition of PVT-vmPFC projections drive and decrease hyperalgesia, respectively. Is PVT driving hyperalgesia through a separate vmPFC node? Similar to Figure 1 a CNO-only control is missing from the hM4Di experiment. Thus, it is unclear if silencing PVT input to the vmPFC opposes or simply masks CFA-induced hyperalgesia.

Just as the reviewer noticed, our results indicate that nNOS-expressing neurons in vmPFC are stimulated by the pain signal from PVT and then pain signal was transduced to anxiety signal.

there is a separation between chronic pain and chronic pain-induced anxiety. Since PVT-vmPFC projections are involved in hyperalgesia and vmPFC nNOS-expressing neurons are not, it is likely that nNOS⁻ neurons in vmPFC are responsible for the PVT driving hyperalgesia. This is demonstrated indirectly by the experiment added in the revision (please see Supplementary Figure 4c, chemogenetic stimulating nNOS⁻ neurons induced a obvious tendency of hyperalgesia). Regarding the partial analgesia mediated by PVT-vmPFC projections inhibition (Figure 3m and Supplementary Figure 8c), we suppose that the perception and integration of chronic pain need other more important structure, such as ACC. This issue had been discussed in the manuscript (page 23, line 20-page 24, line 5).

To address the issue of lacking CNO-treated hM4Di control group in Figure 2l-p of original manuscript (Figure 3m-o in revision), we added an experiment in the revision to observe whether chemogenetic silence of PVT input to vmPFC alone is analgesic (Supplementary Figure 7). As the data shown, this manipulation did not change the withdraw threshold of mice in response to von Frey monofilaments. Therefore we excluded the possibility that silencing PVT input to the vmPFC simply masks CFA-induced hyperalgesia.

7) In Figure 3 the authors look at cFos expression to determine if more nNOS⁺ neurons are active following activation of PVT input with hM3Dq and CNO than in the control (NS) group. The authors should also show what is the overall increase in cFos expression in the vmPFC following PVT input activation and what proportion of these neurons are actually nNOS⁺. Also, it appears that the lack of effect in Figure 3m is inconsistent with data from Figure 2. Can the authors comment on this?

According to the reviewer's suggestion, we quantified the c-Fos⁺ cells in the revised manuscript (please see Figure 4b and 4d; text on page 11, line 7-10). The c-Fos⁺ cells in vmPFC of hM3Dq-stimulated nNOS-Cre mice were more than control, and silencing nNOS neurons in CFA-treated mice reduced c-Fos⁺ number. Instead of the proportion of c-Fos⁺ neurons co-labeled with nNOS⁺, we showed the proportion of nNOS⁺ neurons co-labeled with c-Fos⁺ (Figure 4a and 4c). The later might be more appropriate to illustrate the key role of nNOS-expressing neurons in chronic pain-induced anxiety because it markedly increased after CFA treatment or PVT-vmPFC input activation. However, the proportion of c-Fos⁺

neurons co-labeled with nNOS⁺ was always about 80%, and the change was not obvious.

Figure 3m (Figure 4m in revised manuscript) showed that chemogenetic silence of vmPFC nNOS neurons could not oppose hyperalgesia induced by PVT-vmPFC activation. Figure 2e (Figure 3e in revised manuscript) showed that chemogenetic stimulation of PVT-vmPFC induced hyperalgesia. Figure 2l (Figure 3m in revised manuscript) showed that chemogenetic inhibition of PVT-vmPFC input reversed CFA-induced hyperalgesia. So the lack of effect in Figure 3m (Figure 4m in revised manuscript) indicated that vmPFC nNOS neurons were not involved in hyperalgesia driving by PVT-vmPFC inputs.

8) It isn't totally clear why the authors opted to measure sEPSCs as opposed to mEPSCs to assess glutamatergic transmission in vmPFC. Though the authors show that CFA treatment is associated with increased amplitude of sEPSCs in the vmPFC, it is possible that the effect is mediated by increased network activity. The authors should measure mEPSCs to investigate whether AMPA-mediated CFA-induced plasticity can be observed electrophysiologically. Other measures of AMPA-plasticity such as AMPA/NMDA ratio would be extremely valuable. Also, having assessed this, the authors need to specifically manipulate nNOS⁺ neurons during CFA to investigate whether nNOS⁺ neuronal activity is required for the putative glutamatergic plasticity.

We really appreciate the reviewer's professional comments on the electrophysiological experiments. To our knowledge, Fatt and Katz (*J Physiol.* 1952 May;117(1):109-28) originally referred to miniature synaptic potentials at the frog neuromuscular junction as "miniature spontaneous end plate potentials", and the subsequent terminologies spontaneous EPSCs and miniature EPSCs are often used interchangeably. The two events resemble one another in both amplitude and time course. However, the arrival of pre-synaptic action potentials raises the probability of quantal release of neurotransmitter in sEPSCs recording. So we agree with the reviewer, not only AMPAR plasticity contributes to increased amplitude of sEPSCs in the vmPFC, there is also a possibility that increased amplitude of sEPSCs is mediated by increased network activity. To further confirm the contribution of AMPAR plasticity, we measured AMPA/NMDA ratio as the reviewer's suggestion in the revised manuscript. The results showed that either CFA treatment (Figure 5d-e, text on page 14, line

3-4) or chemogenetic stimulation of nNOS-expressing neurons in vmPFC (Figure 5n-o, text on page 14, line 21-22) obviously increased AMPA/NMDA ratio. We hope these data would strengthen the results from sEPSCs to support the key role of AMPARs. More importantly, we also included an additional experiment in the revised manuscript to investigate whether nNOS⁺ neuronal activity is required for the putative glutamatergic plasticity (Figure 6, text on page 15, paragraph 2). In this experiment, mEPSCs instead of sEPSCs were recorded to exclude the influence from network activity (Figure 6b-d), and AMPA/NMDA ratio were measured too (Figure 6e-f). The results showed that chemogenetic silencing vmPFC nNOS-expressing neurons could reverse CFA-induced increase of mEPSCs amplitude and AMPA/NMDA ratio.

9) The authors provide evidence that chemogenetic activation of nNOS⁺ neurons in vitro, increases sEPSC amplitude and suggest that CFA likely engages similar “cellular” mechanisms. However, if CFA and the chemogenetic activation of nNOS⁺ neurons act through similar mechanisms, then CFA treatment should occlude the effect of the in vitro chemogenetic manipulation. The authors must investigate this experimentally.

We agree with the reviewer that we should better investigate if CFA and the chemogenetic activation of nNOS⁺ neurons act through similar mechanisms with another experiment. However, we do not think it is essential. Based on the data in Figure 5a-r and 6a-f, especially the data in Figure 6a-f which were from an additional experiment performed as the reviewer’s suggestion, we have demonstrated that CFA-induced AMPAR plasticity is dependent on vmPFC nNOS⁺ neurons activity. In other words, activation of nNOS⁺ neurons are the downstream of CFA treatment. So CFA treatment and activation of nNOS⁺ neurons act through same mechanisms certainly. Anyway, we are sorry to have different opinion with the reviewer.

10) Similar to previous experiments where a CNO-only control was missing for hM4Di manipulations, the authors did not include an important control for data shown in Fig. 6d-h. In this figure, the authors show that an NOS specific inhibitor L-VNIO rescues the anxiogenic behavior induced by CFA, but the authors do not include an L-VNIO-only control to

determine whether this manipulation alone is sufficient to alter affective behavior. This needs to be addressed experimentally.

To address this issue, we have included an additional control experiment in the revised manuscript (Supplementary Figure 12, text on page 18, line 19-21). L-VNIO or NS was microinjected into vmPFC through an implanted cannula at day 3 after NS injection within hindpaw planta, and anxiety-related behaviors were tested. Unlike in CFA-treated mice, local administration of L-VNIO in vmPFC did not produce significant anxiolytic effect in NS-injected mice. This finding is consistent with previous study performed in rats restraint stress model with another nNOS inhibitor NPLA (Vila-Verde et al., *Neuroscience*. 2016 Apr 21;320:30-42)

Reviewer #2

Major Concerns

1. My biggest concern of the paper is related to the story of neural circuitry as currently described in the paper. To be clear, the experiments paint an amazing picture of how the PVT could influence nNOS-expressing neurons during chronic pain, causing downstream NO signaling in CaMKII-expressing excitatory neurons in layer II/III mPFC which may induce a state of anxiety that is commonly associated with chronic pain. However, it took me the entire paper to figure out that this is the exact circuitry that the authors found. In my opinion, the authors need to clearly lay out the circuitry in the Abstract, Introduction, discuss how it was tested, and discuss how the circuitry is likely working to control chronic pain-induced anxiety. Additionally, in the Results the authors need to be much clearer about what each particular experiment means for the overall circuit involved. A figure at the end showing PVTmPFCnNOSPFCaMK2a+/nNOS- is something they could consider. The first few sentences of the discussion should also more clearly describe the full circuit that I just mentioned.

First of all, we are deeply grateful to Prof. Otis's kindness in expressing his willingness to

provide us additional assistance with the manuscript writing. Also, we are sorry for our poor writing in English. Although we have done our best to improve writing in the revised version, there must be many problems in language owing to our limited English proficiency. We are looking forward to Prof. Otis's assistance.

According to Prof. Otis's suggestion, we have laid out the circuitry "PVT excitatory neurons / vmPFC nNOS-expressing neurons / NO-mediating AMPARs trafficking in vmPFC pyramidal neurons" in the Abstract of the revision (please see page 3, line 6-8). Meanwhile, we have tried to describe what we would test at the beginning of each experiment and drew a brief conclusion after each experiment. Most importantly, we illustrated the the whole neural circuitry and molecular mechanism with a model diagram in Figure 9 and referred to it in the Discussion (page 21, line 2). We hope that the model diagram will partially overcome our shortcomings in language.

Minor Concerns

1. My second, more minor concern is related to the control treatments used for behavioral studies. For example, in Figure 2 CNO is not given to a control mouse expressing just mCherry instead of hm3Dq (Fig 2e-i) or hm3Di (Fig 2l-p). CNO is known to have off-target effects, and thus it is important to include a control group to show that behavioral results are not related to CNO treatment alone. I would prefer to have this control group for all experiments, but I suppose adding it to just one, revealing that CNO alone does not influence the studied behaviors, would be sufficient. The authors could also probably indicate that CNO induced opposing behaviors in excitatory DREADD and inhibitory DREADD conditions, suggesting no non-specific effects, but this needs to be very clear in the results sections and possibly even in the discussion.

As Prof. Otis pointed out, CNO have off-target effects and it is important to show that behavioral results are not related to CNO treatment alone. In the revision, we included an additional control experiment to address the issue, and stated in the Results that the behavioral changes were not the non-specific effects of CNO itself (please see Supplementary Figure 9, and text on page 10, line 18-19).

2. Being someone from outside of the chronic pain and anxiety fields, I'm confused as to why CFA and von Frey hair stimulation were used. At the very least, references should be used when they are first brought up to indicate other papers that have validated these methods supporting their use as a model system for anxiety and chronic pain.

We used two chronic pain models in the study. Chronic inflammatory pain was induced by CFA (Koga et al., *Neuron*. 2015 Jan 21;85(2):377-89; Wang et al., *Nat Commun*. 2015 Jul 16;6:7660; Jiang et al., *Eur J Pharmacol*. 2015 Jun 15;757:53-8), and chronic neuropathic pain was induced by SNL (Jiang et al., *Eur J Pharmacol*. 2015 Jun 15;757:53-8, Ji et al., *J Neurosci*. 2017 Feb 8;37(6):1378-1393). Both pain models are widely used and can lead to significant anxiety-related behaviors in rodents (Koga et al., *Neuron*. 2015 Jan 21;85(2):377-89; Wang et al., *Nat Commun*. 2015 Jul 16;6:7660; Corder et al., *Science*. 2019 Jan 18;363(6424):276-281; Ji et al., *J Neurosci*. 2017 Feb 8;37(6):1378-1393). To make our paper more readable, we described this issue in Materials and Methods and referred to previous papers (please see page 25, line 14-15). The literature has validated these methods supporting their use as a model system for anxiety and chronic pain. Moreover, we also referred to these papers in Results when they are first brought up (page 6, line 6-7).

Regarding von Frey hair stimulation, it is a classic method to assess mechanical hyperalgesia (Sandkühler, *Physiol Rev*. 2009 Apr;89(2):707-58), and has a wide utilization in pain and pain-induced anxiety research (Koga et al., *Neuron*. 2015 Jan 21;85(2):377-89; Corder et al., *Science*. 2019 Jan 18;363(6424):276-281). As Prof. Otis suggested, we cited references in the text of revised manuscript (Results on page 6, line 6-7 and Materials and Methods on page 25, line 21-page 26, line 1).

3. I find that the authors often overstep conclusions made regarding neural circuit function for a behavior. For example, "...mice exhibited mechanical hyperalgesia..." does not seem entirely supported by the experiments. Instead, the mice expressed behaviors that are consistent with hyperalgesia, or, the experiments suggest that mice were expressing mechanical hyperalgesia. I don't believe the authors have the data to know how the animals felt, unless I'm missing something about how algesia is monitored. The same is true for

conclusions made about neural circuit function for the behaviors studied. Just because we as scientists can bidirectionally modify behavior with CNO, and measure Fos activity, does not mean those circuits causally control those behaviors in natural conditions. Be careful to choose phrasing in a way that accounts for these possibilities, by either explaining more thoroughly or adding words like “suggests” or “indicates”, especially in the results section before the final conclusions for the paper are made.

We really appreciate Prof. Otis’s comments. We should be careful to choose phrasing when writing scientific paper. In the revised version, “...mice exhibited mechanical hyperalgesia...” has been amended as “...mice exhibited decreased 50% paw withdraw threshold...”, please see page 6, line 8. Moreover, we have checked the whole manuscript and added words “suggests” or “indicates” into the sentences used to draw conclusions.

4. Degrees of freedom should be listed for all analyses. T-tests should be described as independent or dependent before judgment can be made about whether the analyses were performed appropriately or not.

Degrees of freedom (df) for ANOVA had been listed in subscript of F in our original manuscript, and df for T-tests have also been included in the text of the revised version. In this study, all T-tests were unpaired. So they were independent. We have stated this issue in the Materials and Methods (page 27, line 14).

5. The “vmPFC” placement is in both prelimbic and infralimbic. Although infralimbic is historically considered part of vmPFC, most researchers do not consider prelimbic to be a part of vmPFC (see Laubach et al., 2018 eNeuro; PMID: 30406193 for a descriptive analysis of this). Thus, the authors likely need to take out the “v” in “vmPFC”, unless they can provide further explanation and citations for why they feel vmPFC is the most appropriate nomenclature. The two citations currently listed (ref #14, 15 in text), are not really appropriate for this claim and should be removed.

We strongly agree with Prof. Otis that there is confusion and controversy about what defines prefrontal cortex and its subregions in rodents. The researchers were evenly split on what areas comprise the dmPFC and vmPFC, especially how to characterize PL. After much

deliberation, we decided to remain the term “vmPFC” in the revised manuscript. We are sorry to have different opinion with Prof. Otis. (1) There is no precise definition for vmPFC. As Laubach et al. reported, 24 of 36 respondents said PL is part of dmPFC and 16 of 36 said it is part of vmPFC (six respondents included PL in both dmPFC and vmPFC). Neither of them is wrong. However, when researchers need a term to indicate both PL and IL, they usually prefer “vmPFC”, including Greg Quirk’s Lab (Figure 1 in *Neuron*. 2007 Mar 15;53(6):871-80; Figure 1 in *Eur J Neurosci*. 2006 Sep;24(6):1751-8; Abstract in *Neuropsychopharmacology*. 2011 Jan;36(2):529-38; Abstract in *Brain Struct Funct*. 2007 Sep;212(2):149-79; Figure 1 in *Biol Psychiatry*. 2014 Aug 1;76(3):203-12; Figure 3 in *Neuroscience*. 2011 Oct 13;193:259-68). (2) The most important reason for us to choose the term “vmPFC” is to exclude the more dorsal and caudal area of PFC, which is called anterior cingulate cortex (ACC) in a series of significant studies about chronic pain and anxiety from Prof. Zhuo’s lab (Zhao et al., *Neuron*. 2005 Sep 15;47(6):859-72; Li et al., *Science*. 2010 Dec 3;330(6009):1400-4; Koga et al., *Neuron*. 2015 Jan 21;85(2):377-89). ACC and mPFC can be used to describe the same cortical regions (Laubach et al., 2018 eNeuro; PMID: 30406193), If we use “mPFC” instead of “vmPFC”, it will cause more confusing, because we cited these “ACC” studies in our manuscript. (3) We had given out stereotactic coordinates and shown the representative coronal section, which, we think, would help the readers to know exactly where the experiment was done.

Additionally, we have removed the citation (ref #14 in text of original manuscript) as Prof. Otis suggestion. However, we retained ref #15 (Hoover and Vertes, *Brain Struct Funct*. 2007 Sep;212(2):149-79) in the revised manuscript. It claimed vmPFC is composed of PL and IL in abstract and text, indeed. Meanwhile, we added another two citations (Veerakumar et al., *Biol Psychiatry*. 2014 Aug 1;76(3):203-12; Kiselycznyk et al., *Neuroscience*. 2011 Oct 13;193:259-68). Their authors regarded vmPFC as PL + IL, and showed this in the figures. Moreover, we modified the sentence “..., which is composed of prelimbic cortex (PL) and infralimbic cortex (IL)...” as “..., which is considered to comprise prelimbic cortex (PL) and infralimbic cortex (IL) by some researchers...” (page 4, line 8-9).

6. Related to the comment above, further clarification about placements would be useful. The

injection site, which is at -2.2DV from Bregma, should hit prelimbic but not infralimbic according to the Paxinos and Watson mouse brain atlas – especially considering that viruses spread mostly upwards, not downwards, after injection. Have the researchers confirmed that both prelimbic and infralimbic cortices have been targeted for all experiments?

We have to admit that we confirmed the placements we targeted in each animal only in the first few experiments. After we were skilled and almost all animals were correctly targeted in both PL and IL, we only confirmed the suspected ones and eliminate the the values from the mice not been correctly targeted.

Regarding the injection depth, we are so sorry for the mistake, the true parameter used in our study is -2.8DV from Bregma. We have corrected it in the revised manuscript (please see Supplementary Information, page 5, line 14).

7. The usefulness of Figure 2D and 2K are limited, given that CNO is infused at the terminal projections, not cell bodies. If the terminals do not contain the G-protein machinery required for CNO to work, then somatic but not terminal inhibition would work. Regardless, if CNO only has a behavioral effect in DREADD-injected animals only (see Major Concern #1), this would mitigate this concern substantially.

We agree with Prof. Otis that the usefulness of Figure 2D and 2K (Figure 3d and 3i in the revision) are limited. Because we microinjected DREADD within the area where cell bodies distributed, it is easier to manipulate the bodies than the terminals. Although somatic inhibition by CNO is not sufficient to guarantee the terminal effects, it is necessary. In other words, terminal inhibition would not work, if somatic inhibition did not work. So, we would like to still include these two data in the revised manuscript, as side evidences. Moreover, we have demonstrated that CNO did not have a behavioral effect without DREADD injection (Supplementary Figure 9), according to Prof. Otis' comments.

8. CAPON should be both described and given full name on page 4 before abbreviation. i.e., what is a partner protein?

CAPON is a cytoplasmic protein whose C terminus binds to the PDZ domain of nNOS, and is designated as CAPON (a carboxy-terminal PDZ ligand of nNOS) by its finders (Jaffrey et

al., *Neuron*. 1998 Jan;20(1):115-24). So we called it a partner protein of nNOS. In the revised manuscript, we have given the full name on page 5, line 4.

9. CFA, NS, SNL, and all other acronyms should be both described and given full names and descriptions before abbreviations.

In the text of revised manuscript, we have given full names for CFA, NS, SNL, GABA, sEPSCs, DREADDs and all other acronyms, and the full chemical names for the abbreviations of compounds, including CNQX, AP-5, L-VNIO, C-PTIO, etc. Moreover, the full chemical names of compounds have also been listed in the Key Resources Table of Supplementary Information.

10. It would be helpful if abbreviations in Figures were listed at end of Figure Legend.

According to Prof. Otis' comment, we tried to list abbreviations in each Figure at end of Figure Legend. However, we found that most abbreviations were repeatedly used in most figures. To save space, we provided a total abbreviation list in alphabetical order in the title page. This list did not include chemical names of compounds, which have been listed in Key Resources Table of Supplementary Information.

11. Some images in the figures are very oversaturated (e.g., Fig 1F and 1J), and thus contrast should be adjusted to allow visualization of the real fluorescence profile of the visualized tissues.

As Prof. Otis pointed out, Fig 1F and 1J (Fig 1d and 1i in the reversion) are oversaturated. The aim was to show the weakly labeled type II nNOS-expressing neurons. We have reduced the contrast of these figures in the revised manuscript. However, the adjustment was limited, or type II nNOS-expressing neurons would not been visualized.

12. Figure 4: C-PTIO is written as C-PITO.

We thank Prof. Otis for his carefullness. We have revised this mistake, please see Figure 5 in the revision.

13. Figure 4: A C-PTIO control group that does not receive CNO should have been included, or the conclusion cannot be made that the elevation in sEPSC amplitude is NO dependent.

We agree with Prof. Otis, a C-PTIO group that does not receive CNO should be included to strengthen our conclusion. So we did an additional self-control experiment to determine whether the amplitude of sEPSCs reduced after C-PTIO treatment. The result revealed that sEPSCs amplitude only have a slight tendency to decrease without significance. We included these new data together with our original data, but only showed the section after C-PTIO treatment as C-PTIO group (Figure 5k-m). Moreover, we measured AMPA/NMDA ratio to further demonstrate the enhanced AMPARs function in the revision. In this experiment, four groups (NS, C-PTIO, CNO and CNO+C-PTIO) were included (Figure 5n-o, text on page 14, line 20-page 15, line 3). Therefore, we concluded that augmentation of vmPFC AMPARs function after activation of nNOS-expressing neurons was dependent on NO production. Additionally, Why did NO scavenger show obvious effect in CNO-treated slices but not NS-treated slices? We attributed the discrepancy to high sensitivity of high NO level after nNOS-expressing neurons were chemogenetically stimulated. In other words, the effect of C-PTIO was dependent on nNOS enzyme activity. this discrepancy also existed in the behavioral result from nNOS inhibitor (Supplementary Figure 12), which indicated that local administration of L-VNIO (a nNOS inhibitor) in vmPFC did not produce significant anxiolytic effect in NS-injected mice, unlike in CFA-treated mice.

14. Figure 4 Legend: It would be helpful if the authors could indicate that the trafficking of AMPARs is in layer II/III excitatory pyramidal neurons as in Figure 5 legend.

According to Prof. Otis's suggestion, we have indicated that the trafficking of AMPARs is in layer 2/3 pyramidal neurons in Legend of Figure 4 (Figure 5 in the revision). Please see page 40, line 7.

15. Sentences and Headings such as “nNOS-expressing neurons in vmPFC mediates chronic pain-induced anxiety through the nNOS-NO pathway” could be somewhat misleading. Make it clear that this is happening through NO signaling in downstream CaMK2a-expressing neurons, not specifically in the nNOS-expressing neurons.

We are sorry for the misleading sentences and headings. In the revised manuscript, “nNOS-expressing neurons in vmPFC mediates chronic pain-induced anxiety through the nNOS-NO pathway” has been modified as “nNOS-expressing neurons in vmPFC mediates chronic pain-induced anxiety through diffusing NO” (please see page 18, line 6). We did not use “CaMKII α -expressing neurons” in this heading directly, because we could not distinguish this neurons population from others in the experiments under this heading (Figure 8). We could draw the nNOS⁺ neurons / diffuding NO / CaMKII α ⁺ neurons pathway only based on the comprehensive analysis of the results from Figure 5 to Figure 8. So we illustrated this neurons-crosstalking pathway in the summary of the whole study by a model diagram in Figure 9. Moreover, the word “diffusing” has been added as an adjunct to the word “NO” in many other sentences to emphasize that the downstream signaling was not specifically in nNOS-expressing neurons.

16. Much of the discussion seems to be reiteration, even at verbatim (see paragraph 2), of what was said in the introduction. Discussion should extend, rather than reiterate, what has been said in the introduction.

To address this issue, we have rewritten paragraph 2 of Discussion and deleted the reiteration. Instead, PFC outputs that have been implicated in anxiety, fear, and motivation has been discussed in the revised manuscript (page 21, line 11-page 22, line 15).

17. This is more of personal preference since I'm a circuit neuroscientist, but I think the authors could spend more time discussing the circuits involved. For example, the different PVT projections that have been implicated in anxiety, fear, and motivation (PVTCEA, PVTPFC, PVTNAc; Greg Quirk Lab, Bo Li Lab, Stuber Lab, Xiaoke Chen Lab). I would also really like to hear more about how PFC outputs from these layer II/III CamK2 neurons might influence those anxiety states. For example, PFCNAc, PFCPVT, PFCBLA circuits have been implicated in anxiety, fear, and/or motivated behaviors (see the same labs, also Peters, Kalivas, Quirk 2009 Learning and Memory for review).

According to Prof. Otis's comments, we have discussed the the different PVT projections and PFC outputs that have been implicated in anxiety, fear, and motivation in paragraph 3 (page

22, line 16-page 23, line 1) and paragraph 2 (page 21, line 11-page 22, line 15) of Discussion, respectively. We thank Prof. Otis for the references recommended. The findings from the studies on mPFC circuits are complex, and it is not easy to speculate which output from mPFC is most likely to mediate chronic pain-induced anxiety. One prominent theoretical perspective highlights the top-down inhibition of amygdala by vmPFC as a crucial neural mechanism that may be defective in certain mood and anxiety disorders (Ball et al., *Psychol Med.* 2013 Jul;43(7):1475-86; Adhikari et al., *Nature.* 2015 Nov 12;527(7577):179-85). Our results, if enhanced AMPARs function necessarily leads to strengthened vmPFC outputs, are contradictory to the defective top-down inhibition, but in agreement with the classic pharmacological studies (Saitoh et al., *J Neurosci Res.* 2014 Aug;92(8):1044-53; Bi et al., *Neuropharmacology.* 2013 Sep;72:148-56; Solati et al., *Acta Neuropsychiatr.* 2013 Aug;25(4):221-6).

18. There are some easily-fixed grammar and spelling errors that should be addressed before publication.

We have checked the whole manuscript carefully and corrected all the grammar and spelling errors we found.

Reviewer #3

This concept is not particularly novel, as the role of the prefrontal cortex in responses to stress or pain-related anxiety has been reported by many investigators. Most of these studies, are not cited in this manuscript. The novelty here involves the focus on specific nNOS cellular populations, whereas the proposed molecular mechanism further supports the well characterized synaptic adaptations in the glutamatergic system and the cell surface expression of AMPA receptors.

We thank the reviewer for his affirmation to our work on specific nNOS cellular populations and the underlying molecular mechanism. As the reviewer pointed out, the role of the prefrontal cortex in responses to stress or pain-related anxiety has been reported by many investigators. So we cited some significant studies of these reports and discussed the complex

mPFC circuits implicated in anxiety, fear, and motivation in the revised manuscript (please see page 21, line 11-page 22, line 15).

The characterization of NOS expressing neurons and circuit analysis involve an early time point after the induction of peripheral inflammation. First of all, I am concerned about the selection of the CFA model for the study of pain-induced anxiety. This peripheral inflammation model leads to spontaneous recovery from sensory hypersensitivity within a few weeks, whereas nerve injury models promote persistent pain states and long-term anxiety. It is surprising that the investigators only observe short term anxiety, as in murine models of chronic pain anxiety-depression like behaviors persist after recovery from mechanical allodynia. Additional models should be used (findings using the SNL model are only reported in supplementary information) to better characterize this circuit. The behavioral analysis is limited to two tests for anxiety-related behaviors, open field and EPM, and to the use Von Frey assays for the assessment of mechanical allodynia. In the current model, mechanical allodynia is observed for several weeks, whereas anxiety is only observed during the first two weeks after CFA treatment. If the goal of the study is to highlight the role of the particular PVT-PFC circuit on pain-induced anxiety, additional behavioral assays should have been included. A wider range of behavioral paradigms reflecting sensory and affective symptoms, as well as motivational states are required in order to better assess the impact of the PFC NOS neurons and the PVT-PFC projections on chronic pain symptomatology.

We agree with the reviewer that CFA model is a peripheral inflammation model leading to spontaneous recovery from sensory hypersensitivity within a few weeks, whereas nerve injury models promote persistent pain states and long-term anxiety. Maybe CFA model is not the perfect model for the study of pain-induced anxiety, it is actually validated to be able to induce anxiety-related behaviors in rodents at days to weeks after CFA injection and is also widely used in this field (Koga et al., *Neuron*. 2015 Jan 21;85(2):377-89; Wang et al., *Nat Commun*. 2015 Jul 16;6:7660; Wang et al., *Mol Brain*. 2019 Apr 8;12(1):36. Jin et al., *Pain*. 2019 Oct 18. doi: 10.1097/j.pain.0000000000001724. [Epub ahead of print]). Another important reason for us to choose CFA model is that inflammatory pain is the most common chronic pain in clinic and inflammation especially neuroinflammation contributes the

induction and maintenance of chronic pain (Ji et al. *Science*. 2016 Nov 4;354(6312):572-577). Additionally, CFA model is easier to prepare than SNL model. Therefore we used CFA model in our most experiments. To confirm our findings from CFA model, we also used SNL model to repeat some important experiments. In the revised manuscript, we removed these results from the Supplementary Information to the main body (please see Figure 2e-h and Figure 8g-i).

Regarding the time points we observed pain-induced anxiety, they were day 3 and day 10 after CFA injection. Although we did not find the anxiety-related behaviors at the last time point day 28, the duration of CFA-induced anxiety might be longer than just 10 days. To address this issue, we added a time point (day 20) in the revision to investigate whether CFA-induced anxiety continued to exist. The result showed that anxiety-related behaviors were still significant though mechanical hyperalgesia became attenuated (please see Supplementary Figure 1g-i and text on page 6, line 15-17). So, CFA-induced anxiety persisted at least 20 days under our experimental condition, consistent with the previous studies on chronic pain-induced anxiety with either CFA or SNL model (Koga et al., *Neuron*. 2015 Jan 21;85(2):377-89; Wang et al., *Nat Commun*. 2015 Jul 16;6:7660; Wang et al., *Mol Brain*. 2019 Apr 8;12(1):36).

As the reviewer mentioned, we used two tests (OF and EPM) for anxiety-related behaviors and one test (von Frey) for mechanical allodynia. OF and EPM are two classical behavioral tests for animal anxiety, which are well-established and broadly-accepted. Most studies even those on top level usually performed only these two tests to assess the anxiety state (Adhikari et al., *Nature*. 2015 Nov 12;527(7577):179-85; Li et al., *Cell*. 2016 Sep 22;167(1):60-72; Koga et al., *Neuron*. 2015 Jan 21;85(2):377-89; Wang et al., *Nat Commun*. 2015 Jul 16;6:7660). We agree that additional behavioral assays will strengthen our conclusions. However, we don't think it is necessary to repeat all experiments, since no additional conclusion we can draw from these repeated data. It will take us more than half a year to repeat all behavioral tests and almost 400 C57BL/6 mice and 200 nNOS-Cre mice will be used. It will be a heavy burden for us. Therefore, we chose two experiments to repeat with novelty-suppressed feeding (NSF) and light-dark box (LDB) test. Please see Figure 1n-o and text on page 7, line 20-22 for confirming the role of vmPFC nNOS-expressing neurons, and

Figure 3i-j and text on page 10, line 1-3 for confirming the role of PVT-vmPFC inputs. The results are consistent with those obtained from OF and EPM. Meanwhile, we measured thermal hyperalgesia in these two added experiments by heat stimuli (radiant heat source) instead of mechanical hyperalgesia to evaluate the chronic pain state (Figure 1p and Figure 3h). The results were similar to those from von Frey test. We hope that these two additional experiments would partially address the reviewer's concern about behavioral tests for pain and anxiety.

Finally, we appreciate the reviewer's comment on a wider range of behavioral paradigms reflecting sensory and affective symptoms, as well as motivational states. Recently an outstanding research like this was published in *Science* (Corder et al., An amygdalar neural ensemble that encodes the unpleasantness of pain. *Science*. 2019 Jan 18;363(6424):276-281). We cited this paper in the Discussion of the revised manuscript (page 22, line 7). This is an interesting and significant field associated with chronic pain. We would like to do a wider range study to better assess the impact of the mPFC nNOS neurons and the PVT-mPFC projections on chronic pain symptomatology in the future, if we got new funding. We are so sorry we could not perfectly complete all these studies in current paper.

Other comments

-Western blot analysis studies: Figures show representative blots, a larger amount or samples or the full membrane along with loading controls should be provided.

According to the reviewer's suggestion and the requirement of the journal, the full memberane along with loading controls has been provided in the Source Data file. For the reviewer's convenience, we pasted the copy in this letter. Please note that the bolts of GAPDH in Supplementary Figure 14 (Figure 7 in the 1st version) were replaced by those from the same experiment of the same tissue samples. In the 1st version, the GAPDH blots were from the replicated experiment of the same tissue samples. When organizing the Source Data file, we found this mistake and corrected it. We are very sorry for our carelessness.

Figure 5f

Figure 5p

Figure 8a

Supplementary Figure 14a

Supplementary Figure 14d

-Several statements are confusing, mostly due to language issues.

We are sorry for our poor writing in English. Although we have done our best to improve writing in the revised version, there must be some problems in language owing to our limited English proficiency. Fortunately, reviewer #2 expressed his willingness to provide us additional assistance with the manuscript writing. We believe that substantial improvement will be achieved after the polishing.

Reviewers' Comments:

Reviewer #1:

Remarks to the Author:

In revising their manuscript, Prof. Lou and colleagues conducted a series of additional controls, manipulations and analyses that significantly improved the quality of the original version. Overall, these additions support the initial model that nNOS+ neurons of the mPFC transduce pain signals, arriving in part from the PVT, into anxiety signals. Also, from a technical perspective, the work is quite profound. As such, it is this reviewer's opinion, that the work is novel and detailed enough to merit publication in Nat. Comm. However, and while I appreciate the effort that the authors have placed in the revision of their manuscript, I consider that several important details still need to be addressed

- 1) The authors provide new immunofluorescence and ISH data that seemingly serve to validate the nNOS-Cre line. The sample images provided all look great. However, no quantification has been provided in the figure legend (Supplementary Figure 3) or the main text.
- 2) For the Cre- neurons experiment in Supplementary Figure 4, there seems to be some effect of the CNO in open field and a clear trend in the von Frey score. Because for the OFT the variability in the distribution seems much bigger than the vehicle group, one can't help it but wonder if this could be due to variability in the infection rate of the hM3q vector. Actually, there are no reports of infection rate here. This is critical information, since one can't know for sure what the distribution and density of hM3q+ cells looks like across animals. The authors should add this information for each one of their DREADD experiments.
- 3) As mentioned above, there are also mild effects of activating Cre- neurons on allodynia (hyperalgesia, in this case). The data appears to show an almost significant effect. What is the p value here? Also, can the authors plot the behavior (von Frey) as a function of the infection rate per subject? It could be that the Cre- neurons might be important for hyperalgesia (contrary to the Cre+), as the authors suggest in the rebuttal. A detailed discussion of this in the manuscript would be valuable for the reader, as it may provide evidence for how the mechanical hyperalgesia and the affective components of pain may be segregated in the mPFC.
- 4) The idea that inhibition of Nos+ neurons with CNO alone doesn't generate anxiolytic effect is not entirely convincing based on Supplementary Fig 5. Again, unlike other data in the paper, p values for OFT result is not provided in the supplementary material. Seems like there is an anxiolytic effect here, which the authors call non-significant. Could differences in infection rate be at play here? The authors should address this.
- 5) The authors claim to have targeted the mid-PVT, but the image looks like posterior PVT. This is critical information, because studies linking the PVT to pain have mainly suggested a role for the anterior (not the posterior) PVT. The authors should discuss the potential relevance of this regional differences. For reference, see a study recently published in Nature Neuroscience (Gao et al., 2020).
- 6) Based on the authors' findings, the PVT-mPFC pathway seems to contribute to (not mediate) hyperalgesia, since modulating this projections only partially rescues the behavior. Caution must be used when interpreting this result. This hyperalgesia could be mediated by PVT input onto nNOS-neurons, as mentioned above.
- 7) For the cfos experiment in vmPFC following chemogenetic activation of PVT, I do not like that the authors only show the proportion of NOS+ cells that also have cfos. While that is relevant, what is equally important is to determine what fraction of the activated mPFC neurons, NOS+ cells represent.

Reviewer #2:

Remarks to the Author:

The authors did a thorough job of addressing my comments and criticisms. Although I am outside of the pain field, I find the circuit experiments to be well performed, novel, and very insightful. At this time, I think the manuscript could be accepted with only minor edits.

Minor comment: I am not entirely satisfied with the abstract, and particularly the new sentence at the end. I think this is a really important part of the authors' manuscript, since the circuitry and experiments are quite complex and therefore it is quite difficult to convey such complex ideas into a condensed paragraph. Since the authors seem open to help with the writing, I took a shot at providing my minor edits to the abstract which the authors could consider using if they find such edits to be accurate and helpful:

"Anxiety is common in patients suffering from chronic pain, and how chronic pain induces anxiety is still not fully understood. Here, we report anxiety-like behaviors in mouse models of chronic pain, and using optogenetic and chemogenetic strategies we reveal that nNOS-expressing neurons in ventromedial prefrontal cortex (vmPFC) are essential for pain-induced anxiety but not algesia. Additionally, we determine that excitatory projections from the paraventricular nucleus of the thalamus (PVT) provide a neuronal input that drives the activation of vmPFC nNOS-expressing neurons in our chronic pain models. Our results suggest that the pain signal is transduced to an anxiety signal after activation of vmPFC nNOS-expressing neurons which causes subsequent release of nitric oxide (NO) that is responsible for chronic pain-induced anxiety. Finally, we show that the downstream molecular mechanisms from NO are likely enhanced glutamate transmission in vmPFC CaMK2a-expressing neurons through S-nitrosylation-induced AMPA receptor trafficking. Overall, our data suggest that PVT excitatory neurons drive chronic pain-induced anxiety through activation of vmPFC nNOS-expressing neurons resulting in NO-mediated AMPA receptor trafficking in vmPFC pyramidal neurons."

Typos: Line 127, 128, 785: Fiugre should be 'Figure'

I do not need to see further edits to the manuscript, as I think it should be accepted with minor revisions at this time.

Best wishes, James Otis

Reviewer #3:

Remarks to the Author:

The authors have provided additional references, and information on western blot analysis. They have added the novelty suppressed feeding assay as a behavioral measure of anxiety-like states. Additional time points have been added for the anxiety assay, and some findings are validated using a nerve injury model. The manuscript has improved, but there are still language issues, and it is confusing to see at several points statements such as "chronic pain is abolished" when they only measure mechanical hyperalgesia. Overall, the characterized circuits may contribute to pain-induced anxiety, but it is a small and not lasting effect.

Reviewer #1

1) The authors provide new immunofluorescence and ISH data that seemingly serve to validate the nNOS-Cre line. The sample images provided all look great. However, no quantification has been provided in the figure legend (Supplementary Figure 3) or the main text.

According to the reviewer's suggestion, we have provided the quantification of IF and FISH in the legend of supplementary figure 3. Please see page 23, line 3 and 7 in Supplementary Information. About 96% mCherry⁺ neurons were nNOS positive for IF and about 92% eGFP mRNA⁺ neurons were nNOS mRNA positive for FISH. These data were obtained from 4 pictures of 4 mice and 10 pictures of 5 mice, respectively. Although they are not precise values from the rigorous quantification with a series of coronal sections of mice, we think these data along with the representative images are sufficient to validate the nNOS-Cre line.

2) For the Cre- neurons experiment in Supplementary Figure 4, there seems to be some effect of the CNO in open field and a clear trend in the von Frey score. Because for the OFT the variability in the distribution seems much bigger than the vehicle group, one can't help it but wonder if this could be due to variability in the infection rate of the hM3q vector. Actually, there are no reports of infection rate here. This is critical information, since one can't know for sure what the distribution and density of hM3q⁺ cells looks like across animals. The authors should add this information for each one of their DREADD experiments.

We appreciate the reviewer's carefulness and meticulousness. As the reviewer pointed out, the variability in the distribution of CNO group was bigger than the vehicle group for the OF test in Supplementary Figure 4. We completely understand the reviewer's concern about the infection rate of the hM3q vector. However, the distribution variability of CNO group was really normal and common, compared with all other OF tests performed in this study, whenever recombinant AAVs were used or not. Just the variability of NS group in Supplementary Figure 4d was fortunately smaller than general ones. Moreover, if the infection rate of the hM3q vector led to the distribution variability of CNO group for OF test, it should have similar influences on the results of von Frey assay and EPM test, as these

behaviors were tested with the same mice. But the distribution variability of data in CNO group was equal to NS group in both von Frey (Supplementary Figure 4c) and EPM (Supplementary Figure 4e). Therefore, we think there is nothing wrong with the infection.

The reviewer suggested us add the infection rate for each one of our DREADD experiments. It is useful to guarantee each data is “true”. However, it is technically impossible to detect the infection rate following each DREADD experiment, because the brains are usually used for other biochemical measurements or electrophysiological recordings after behavioral tests in the formal experiments. So we determined the infection rate of each AAV-DREADD with 2-3 mice before we began the formal experiments. And in the formal experiments, we examined the suspected mice with abnormal data and excluded the values from the mice if the infection rate was lower than 70%. Moreover, we are skilled at microinjection with stereotaxic apparatus and have published a series of work in this field. Correct target would help to increase the infection rate. In the revised manuscript, we provided the infection rate of each AAV-DREADD obtained in our preliminary experiments: DIO-hM3Dq, ~83%; DIO-hM4Di, ~84%; hM3Dq, ~87%; hM4Di, ~83%; DO-hM3Dq, ~81%. Please see page 28, paragraph 2 in text.

3) As mentioned above, there are also mild effects of activating Cre- neurons on algesia (hyperalgesia, in this case). The data appears to show an almost significant effect. What is the p value here? Also, can the authors plot the behavior (von Frey) as a function of the infection rate per subject? It could be that the Cre- neurons might be important for hyperalgesia (contrary to the Cre+), as the authors suggest in the rebuttal. A detailed discussion of this in the manuscript would be valuable for the reader, as it may provide evidence for how the mechanical hyperalgesia and the affective components of pain may be segregated in the mPFC.

Yes, the data in Supplementary Figure 4c show an almost significant effect. $t_{24} = 2.040$, $p = 0.053$. We have included this information in the revision (page 8, line 11 in text, and page 24, line 4 in supplementary information). But we are sorry that we could not provide the plot suggested by the reviewer, as we did not measure the infection rate of each animal for the reason mentioned above. It is a pity. On the other hand, we are confident on our data. The

distribution variability was normal, excluding the possibility of infection variability. Moreover, the variability of behavioral experiments is relatively bigger than other experiments. Even if we plot the behavior (von Frey) as a function of the infection rate per subject, it may not yield a meaningful outcome. High infection rate results in high behavioral effect in theory, but there is not a certain one-to-one match between the behavioral data and the infection rate, regarding the big variability of behaviors itself. Anyway, we thank the reviewer for his agreement with us that the clear trend in Supplementary Figure 4c “could be that the Cre- neurons might be important for hyperalgesia (contrary to the Cre+)”. As the reviewer’s suggestion, we have included a detailed discussion on this issue (page 25, line 12-18 of text). We wish this would be valuable for the reader.

4) The idea that inhibition of Nos⁺ neurons with CNO alone doesn’t generate anxiolytic effect is not entirely convincing based on Supplementary Fig 5. Again, unlike other data in the paper, p values for OFT result is not provided in the supplementary material. Seems like there is an anxiolytic effect here, which the authors call non-significant. Could differences in infection rate be at play here? The authors should address this.

To show the difference between groups in OF test of Supplementary Figure 5, p value ($t_{24} = 1.615$, $p = 0.119$) has been provided in the figure legend (page 24, line 15-17 in Supplementary Information). There is a trend that inhibition of nNOS⁺ neurons with CNO alone generate anxiolytic effect, but the trend is not strong. Because the data distribution variability of CNO group is normal (compared with all other OF tests performed in this study, whenever recombinant AAVs were used or not), the chance that differences in infection rate masked the anxiolytic effect is little. Most importantly, the data of EPM obtained from the same mice did not show any trend of anxiolytic effect (Supplementary Figure 5c, $t_{24} = 0.186$, $p = 0.854$). So, we drew the conclusion that chemogenetic inhibition of nNOS-expressing neurons in vmPFC did not produce significant anxiolytic effect in normal animals.

5) The authors claim to have targeted the mid-PVT, but the image looks like posterior PVT. This is critical information, because studies linking the PVT to pain have mainly suggested a role for the anterior (not the posterior) PVT. The authors should discuss the potential

relevance of this regional differences. For reference, see a study recently published in *Nature Neuroscience* (Gao et al., 2020).

We have read the paper recommended by the reviewer carefully. Gao and her colleagues demonstrate that two genetically, anatomically and functionally distinct cell types segregate across anteroposterior axis of paraventricular thalamus, based on their control of diverse processes such as arousal, stress, emotional memory and motivation. As mentioned in the first paragraph of the Discussion of this paper, the anterior PVT (aPVT) starts at bregma -0.2 mm and the posterior PVT (pPVT) starts at bregma -1.82 mm in mice. However, the authors used sections located at bregma -1.70 mm as pPVT (definition in Fig 1), and some pPVT images in their paper look like more anterior (about bregma -1.58 mm). These locations belong to middle PVT (mPVT) according to *The Mouse Brain in Stereotaxic Coordinates, Third Edition* (Keith B. J. Franklin & George Paxinos, Academic Press of Elsevier). We guess this is why the reviewer felt our mPVT images look like pPVT. The PVT images in Figure 3a and 3c are about at bregma -1.6 mm, and the PVT image in Supplementary Figure 8a is about at bregma -1.3 mm, in our manuscript.

Although the above paper does not suggest any correlation between aPVT and pain, it actually reveals that the function of distinct parts of PVT segregate in controlling arousal and stress. Meanwhile, a series of work from Prof. Chen's lab linking the PVT to pain have mainly suggested a role for the aPVT (*Pain*. 2019 May;160(5):1208-1223 and *J Neurosci*. 2010 Aug 4;30(31):10360-8). In our study, we demonstrate a critical role of mPVT in the chronic pain and the induced anxiety. We have cited these papers and discussed the potential relevance of this regional difference. Please see page 24, line 8 ~ page 25, line 2 in text. Additionally, though Prof. Chen emphasized the role of aPVT, they did not exclude the role of other subregions of PVT in the pain.

6) Based on the authors' findings, the PVT-mPFC pathway seems to contribute to (not mediate) hyperalgesia, since modulating this projections only partially rescues the behavior. Caution must be used when interpreting this result. This hyperalgesia could be mediated by PVT input onto nNOS- neurons, as mentioned above.

We have taken the reviewer's advice and replaced "mediate" by "contribute to" when

interpreting the role of PVT-vmPFC pathway in hyperalgesia in the revised manuscript. Please see page 9, line 8 and page 11, line 5 of text.

7) For the cfos experiment in vmPFC following chemogenetic activation of PVT, I do not like that the authors only show the proportion of NOS⁺ cells that also have cfos. While that is relevant, what is equally important is to determine what fraction of the activated mPFC neurons, NOS⁺ cells represent.

In the revision, we have added the c-Fos⁺&nNOS⁺/c-Fos⁺ ratio in the Figure 4 and described in the text (page 11, line 18-20). Unlike the ratio of c-Fos⁺&nNOS⁺/nNOS⁺, the ratio of c-Fos⁺&nNOS⁺/c-Fos⁺ was not markedly changed when PVT-vmPFC was stimulated (Figure 4c, $t_{10} = 0.206$, $p = 0.841$) or inhibited (Figure 4f, $F_{2,9} = 0.157$, $p = 0.857$) by chemogenetics. Increased c-Fos⁺&nNOS⁺/nNOS⁺ ratio suggests that more nNOS⁺ cells are activated for mediating anxiety when PVT-vmPFC was stimulated. Increased total c-Fos⁺ cells and unchanged c-Fos⁺&nNOS⁺/c-Fos⁺ ratio suggest that nNOS⁺ cells are not preferentially activated, compared with nNOS⁻ cells. Not only nNOS⁺ cells but also nNOS⁻ cells are responsible for PVT-vmPFC stimulation. The latter could be important for hyperalgesia, as we have discussed in the revised manuscript.

Reviewers' Comments:

Reviewer #1:

Remarks to the Author:

In their revised manuscript, the authors have appropriately addressed most of my comments. However, as someone with strong expertise in the PVT, I can assure unequivocally that the images shown in Figure 3 depict the "posterior PVT". Three prominent features observed in the images validate my assessment:

- 1) The volume of the Lateral Habenula (LHb), which is quite large in the images.
- 2) The presence of the fasciculus retroflexus (fr), which separates the habenula and the PVT at posterior-only coordinates.
- 3) The semi-concave shape at the bottom of the PVT, another distinctive feature of the posterior PVT.

At more middle PVT locations, the PVT has a central tail that extends ventrally into the IMD, the fr is absent and the LHb is significantly smaller. Having said this, few studies even recognize this mPVT region. Part of the reason for this is that the boundaries can become hard to identify. However, while for some studies mPVT could be claimed as the main target, based on the histology provided by the authors, I have no doubt that they have targeted the pPVT. As such, I recommend that the authors re-phrase their comments on PVT in the manuscript to refer to it as pPVT. There has been significant confusion in the literature about the antero-posterior PVT location documented in many studies. As a result, I want to do my best to contribute to the rectification of this problem. If the authors agree to correct their statement and refer to pPVT, I would be happy to support publication of their study. I hope that the authors appreciate that my goal here is that their study is as accurate as possible. Their study would be the first to recognize a role for the pPVT in pain, thereby expanding on previous literature demonstrating a role for the aPVT in this process.

Reviewer #1

In their revised manuscript, the authors have appropriately addressed most of my comments. However, as someone with strong expertise in the PVT, I can assure unequivocally that the images shown in Figure 3 depict the "posterior PVT". Three prominent features observed in the images validate my assessment:

- 1) The volume of the Lateral Habenula (LHb), which is quite large in the images.
- 2) The presence of the fasciculus retroflexus (fr), which separates the habenula and the PVT at posterior-only coordinates.
- 3) The semi-concave shape at the bottom of the PVT, another distinctive feature of the posterior PVT.

At more middle PVT locations, the PVT has a central tail that extends ventrally into the IMD, the fr is absent and the LHb is significantly smaller. Having said this, few studies even recognize this mPVT region. Part of the reason for this is that the boundaries can become hard to identify. However, while for some studies mPVT could be claimed as the main target, based on the histology provided by the authors, I have no doubt that they have targeted the pPVT. As such, I recommend that the authors re-phrase their comments on PVT in the manuscript to refer to it as pPVT. There has been significant confusion in the literature about the antero-posterior PVT location documented in many studies. As a result, I want to do my best to contribute to the rectification of this problem. If the authors agree to correct their statement and refer to pPVT, I would be happy to support publication of their study. I hope that the authors appreciate that my goal here is that their study is as accurate as possible. Their study would be the first to recognize a role for the pPVT in pain, thereby expanding on previous literature demonstrating a role for the aPVT in this process.

Many thanks for the reviewer's professional comments. We have referred to the PVT as the pPVT in the final version of the manuscript, including text and figures. Additionally, we have re-phrased our discussion on PVT. For the limiting space in the main text, we removed this paragraph to Supplementary Discussion.